# AVIS: Autonomous Visual Information Seeking with Large Language Model Agent

**Ziniu Hu**[1][2][*]    **Ahmet Iscen**[2]    **Chen Sun**[2]    **Kai-Wei Chang**[1]    **Yizhou Sun**[1]

**David A Ross**[2]    **Cordelia Schmid**[2]    **Alireza Fathi**[2]

[1]University of California, Los Angeles, [2]Google Research

## Abstract

In this paper, we propose an autonomous information seeking visual question answering framework, AVIS. Our method leverages a Large Language Model (LLM) to dynamically strategize the utilization of external tools and to investigate their outputs via tree search, thereby acquiring the indispensable knowledge needed to provide answers to the posed questions. Responding to visual questions that necessitate external knowledge, such as "What event is commemorated by the building depicted in this image?", is a complex task. This task presents a combinatorial search space that demands a sequence of actions, including invoking APIs, analyzing their responses, and making informed decisions. We conduct a user study to collect a variety of instances of human decision-making when faced with this task. This data is then used to design a system comprised of three components: an LLM-powered planner that dynamically determines which tool to use next, an LLM-powered reasoner that analyzes and extracts key information from the tool outputs, and a working memory component that retains the acquired information throughout the process. The collected user behavior serves as a guide for our system in two key ways. First, we create a transition graph by analyzing the sequence of decisions made by users. This graph delineates distinct states and confines the set of actions available at each state. Second, we use examples of user decision-making to provide our LLM-powered planner and reasoner with relevant contextual instances, enhancing their capacity to make informed decisions. We show that AVIS achieves state-of-the-art results on knowledge-intensive visual question answering benchmarks such as Infoseek [7] and OK-VQA [26].

## 1 Introduction

Large language models (LLMs), such as GPT3 [5], LaMDA [16], PALM [9], BLOOM [34] and LLaMA [37], have showcased the capacity to memorize and utilize a significant amount of world knowledge. They demonstrate emerging abilities [38] like in-context learning [5], code generation [19], and common sense reasoning [24]. Recently, there is a growing focus towards adapting LLMs to handle multi-modal inputs and outputs involving both vision and language. Noteworthy examples of such visual language models (VLMs) include GPT4 [29], Flamingo [4] and PALI [6]. They set the state of the art for several tasks, including image captioning, visual question answering, and open vocabulary recognition.

While LLMs excel beyond human capabilities in tasks involving textual information retrieval, the current state of the art VLMs perform inadequately on datasets designed for visual information

---

[*]This work was done when Ziniu was an intern at Google.

37th Conference on Neural Information Processing Systems (NeurIPS 2023).

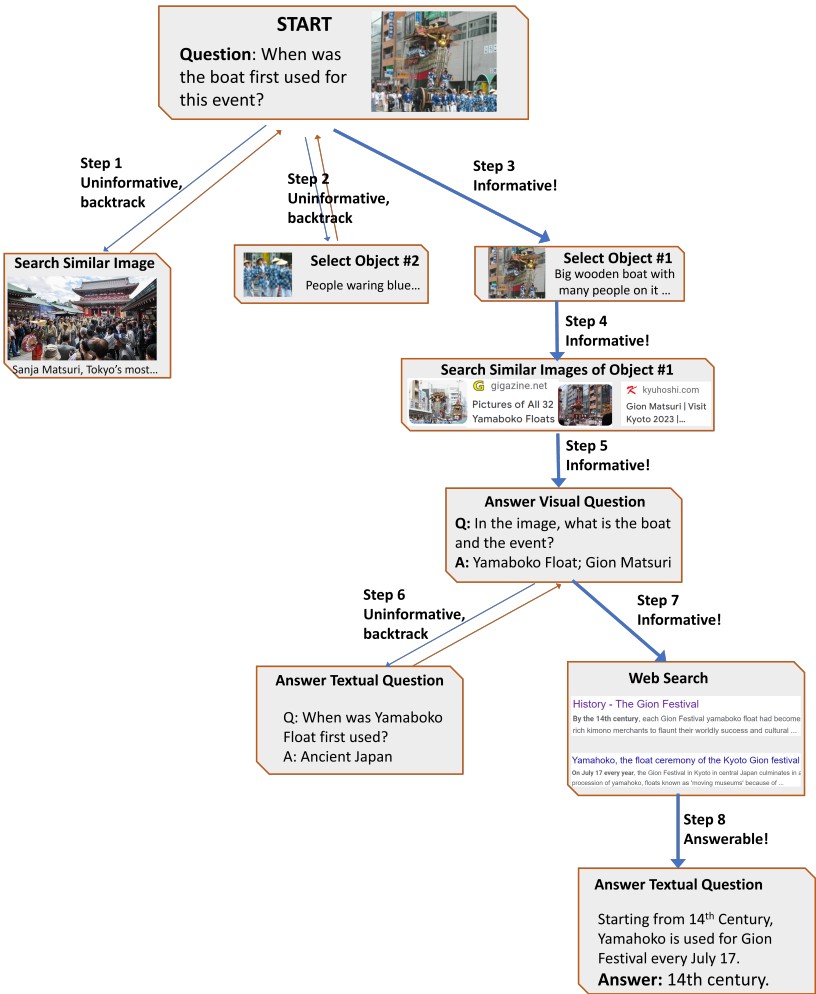

Figure 1: An example of AVIS's generated workflow for answering a challenging visual question using LLM with tree search to use tools. The input image is taken from the Infoseek dataset.

seeking such as Infoseek [7] and OK-VQA [26]. Many of the visual questions in these datasets are designed in such a way that they pose a challenge even for humans, often requiring the assistance of various APIs and web search to obtain the answer. Examples of such questions include "where is this church located?", "what species of butterfly is this?", or "what is the brand of this dress?".

Current state-of-the-art vision-language models (VLMs) find it challenging to answer such questions for several reasons. Firstly, they are not trained with objectives that encourage them to discern fine-grained categories and details within images. Secondly, they utilize a relatively smaller language model compared to state-of-the-art Large Language Models (LLMs), which constrains their reasoning capabilities. Lastly, they do not compare the query image against a substantial corpus of images associated with varying metadata, unlike systems that employ image search techniques.

To overcome these challenges, we introduce a novel method in this paper that achieves state-of-the-art results on visual information seeking tasks by enabling a **LLM Agent use tools via tree-search decision-making**. We use three types of tools: (i) computer vision tools such as object detection, OCR, image captioning models, and VQA models, which aid in extracting visual information from the image, (ii) a web search tool that assists in retrieving open world knowledge and facts, and (iii) an image search tool that enables us to glean relevant information from metadata associated with visually similar images. Our approach utilizes an LLM-powered planner to dynamically determine which tool to use at each step and what query to send to it. Furthermore, we employ an LLM-powered reasoner that scrutinizes the output returned by the tools and extracts the crucial information from them. To retain the information throughout the process, we use a working memory component. Figure 1 shows an example information seeking process performed by our method.

Several recent studies [13, 23, 36, 40, 42] have enhanced LLMs with APIs to handle multi-modal vision-language inputs. These systems generally employ a two-stage strategy, namely *plan* and *execute*. Initially, the LLM breaks down a question into a plan, typically represented as a structured

program or a sequence of instructions. Following this, the necessary APIs are activated to collect the required information. While this method has shown potential in elementary visual-language tasks, it frequently fails in more complex real-world situations. In such cases, a comprehensive plan cannot be inferred merely from the initial question. Instead, it necessitates dynamic modifications based on real-time feedback.

The primary innovation in our proposed method lies in its dynamic decision-making capability. Answering visual information seeking questions is a highly complex task, requiring the planner to take multiple steps. At each of these steps, the planner must determine which API to call and what query to send. It is unable to predict the output of complex APIs, such as image search, or to anticipate the usefulness of their responses prior to calling them. Therefore, unlike previous methods that pre-plan the steps and API calls at the beginning of the process, we opt for a dynamic approach. We make decisions at each step based on the information acquired from previous API calls, enhancing the adaptability and effectiveness of our method.

We conduct a user study to gather a wide range of instances of human decision-making when using APIs to answer questions related to visual information seeking. From this data, we formulate a structured framework that directs the Large Language Model (LLM) to use these examples for making informed decisions regarding API selection and query formulation. The collected user behavior informs our system in two significant ways. First, by analyzing the sequence of user decisions, we construct a transition graph. This graph delineates distinct states and constrains the set of actions available at each state. Second, we use the examples of user decision-making to guide our planner and reasoner with pertinent contextual instances. These contextual examples contribute to improving the performance and effectiveness of our system.

The primary contributions of this paper can be summarized as follows:

- We propose a novel visual question answering framework that leverages a large language model (LLM) to dynamically strategize the utilization of external tools and to investigate their outputs, thereby acquiring the necessary knowledge needed to provide answers to the posed questions.
- We leverage the human decision-making data collected from a user study to develop a structured framework. This framework guides the Large Language Model (LLM) to utilize examples of human decision-making in making informed choices concerning API selection and query construction.
- Our method achieves state-of-the-art results on knowledge-based visual question answering benchmarks such as Infoseek [7] and OK-VQA [26]. Notably, We achieve an accuracy of $50.7\%$ on the Infoseek (unseen entity split) dataset which is significantly higher than the results achieved by PALI [6] with accuracy of $16.0\%$.

## 2   Related Work

**Augmenting LLMs with Tools.**   Large Language Models(LLMs) have shown impressive language understanding [33], and even reasoning capabilities [39]. Nevertheless, certain limitations of LLMs are evident, due to their intrinsic characteristics. Such limitations include providing up-to-date answers based on external knowledge or performing mathematical reasoning. Consequently, a recent surge of techniques have integrated LLMs with various external tools [27]. For example, TALM [31] and ToolFormer [35] use in-context learning to teach the language model how to better leverage various tools on benchmarks such as question answering and mathematical reasoning.

In the computer vision domain, LLMs also show significant improvements when combined with external visual tools. For example, Visual ChatGPT [40] and MM-ReAct [42] enable LLMs to call various vision foundation models as tools to understand visual inputs, and even better control the image generation. VisProg [13] and ViperGPT [36] explore the decomposition of visual language tasks into programs, where each line corresponds to general code or a visual API. Chameleon [23] uses an LLM as a natural language planner to infer the appropriate sequence of tools to utilize, and then executes these tools to generate the final response.

Most of these previous works follow a plan-then-execute paradigm, i.e., i) they pre-plan the sequence of actions (API calls) that they will take (either hard coded or using code generation); and ii) they execute the generated plan. One drawback of such an approach is that it cannot update and improve

its plan based on the output of the tools it calls. This is not a trivial problem, as it requires to predict the output quality of each tools beforehand. In contrast, our proposed method allows the system to dynamically decide its next steps based on the output it receives from the tools at each step.

**Decision Making with LLM as an Agent.**  There has also been a surge of interest in applying Large Language Models (LLMs) as autonomous agents. These agents are capable of interacting with external environments, making dynamic decisions based on real-time feedback, and consequently achieving specific goals. For example, WebGPT [28] enables an LLM to access real-time information from the web search engines. ReAct [44] further improves external search engine usage via the self-reasoning of LLM in an interleaved manner. Similar ideas have also been adopted for robotic action planning. SayCan [3], for instance, uses LLMs to directly predict robot actions, and PALM-E [10] further fine-tunes LLMs to make better decisions based on instructions and open web media.

When compared to works that follow a plan-then-execute paradigm, these AI agents exhibit increased flexibility, adjusting their actions based on the feedback that they receive. However, many of these methods do not restrict the potential tools that can be invoked at each stage, leading to an immense search space. This becomes particularly critical for web search APIs [1, 2] that return extensive result lists and span a combinatorial search space of multiple tools. Consequently, even the most advanced LLMs today can fall into infinite loops or propagate errors. To alleviate this issue, we propose restricting and guiding LLMs to mimic human behavior when solving complex visual questions with APIs. This idea is similar to the AI alignment research [21, 30] that teaches LLMs to follow human instructions. The difference is that our model only uses the human prior at the decision-making stage via prompt guidance, instead of re-training the model.

One concurrent work Tree-Of-Thought (ToT) [43] also utilize tree search guided by a self-critic reward model to find optimal path of problem solving. Compared with this concurrent work, our AVIS further constrains the tree search via a human-defined transition graph, and guide the decision-making via a dynamic prompt manager. In addition, though AVIS is designed for tool-use, the success of ToT shows that such idea can be generally improve many LLM Reasoning tasks.

## 3 Method

### 3.1 General Framework

Our approach employs a dynamic decision-making strategy designed to respond to visual information-seeking queries. Our system is comprised of three primary components. First, we have a planner $\mathcal{P}$, whose responsibility is to determine the subsequent action, including the appropriate API call and the query it needs to process. Second, we have a working memory $\mathcal{M}$ that retains information about the results obtained from API executions. Lastly, we have a reasoner $\mathcal{R}$, whose role is to process the outputs from the API calls. It determines whether the obtained information is sufficient to produce the final response, or if additional data retrieval is required.

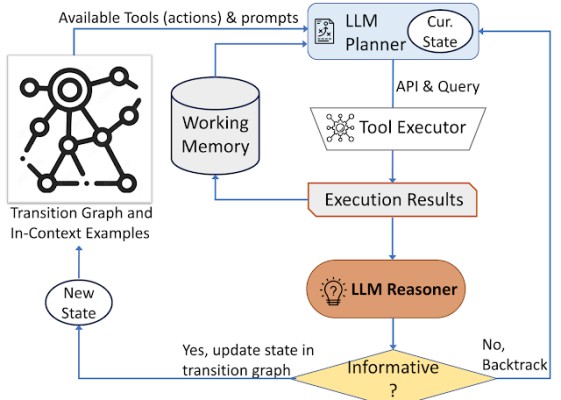

Figure 2: AVIS employs dynamic decision-making to **plan** (find optimal tool and query), execute results, and then **reason** (estimate whether continue or backtrack).

**Algorithm 1** Planner $\mathcal{P}(state, \mathcal{G}, \mathcal{E}, \mathcal{M})$

1: $\mathcal{A}_s \leftarrow \phi(state, \mathcal{G}, \mathcal{M})$  ▷ Get the list of feasible actions $\mathcal{A}_s$ given the current state from transition graph and the information in the working memory
2: $\mathcal{E}_s \leftarrow \theta(\mathcal{E}, \mathcal{A}_s)$  ▷ Get a list of in-context examples related to actions $\mathcal{A}_s$
3: $p_s \leftarrow \psi(\mathcal{E}_s, \mathcal{M})$  ▷ Build a prompt based on the in-context examples $\mathcal{E}_s$ and the current working memory $\mathcal{M}$
4: $t_s, q_s \leftarrow LLM(p_s)$ ▷ Decide the next tool $t_s$ to use and the query $q_s$ to pass by feeding the prompt $p_s$ to LLM

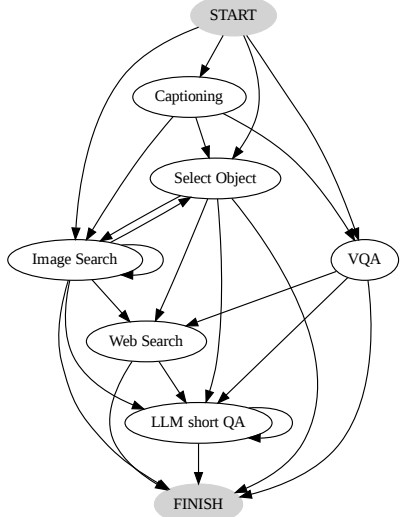

START

Captioning

Select Object

Image Search

VQA

Web Search

LLM short QA

FINISH

Figure 3: Transition graph $\mathcal{G}$ defines feasible actions the planner can take. This graph is induced by our user study introduced in Sec. 3.3.

---

**Algorithm 2** AVIS Decision Making Workflow
1: $\mathcal{M} \leftarrow \{input\}, state \leftarrow \text{START}$
2: $t_s, q_s \leftarrow \mathcal{P}(state, \mathcal{G}, \mathcal{E}, \mathcal{M})$  ▷ Call the planner $\mathcal{P}$ to decide the next tool to use $t_s$ and the query to pass to it $q_s$
3: $o_s \leftarrow \text{Exec}(t_s, q_s)$  ▷ Call tool $t_s$ with query $q_s$ and get output $o_s$
4: $\hat{o}_s \leftarrow \mathcal{R}(o_s, \mathcal{M})$  ▷ Process the output and extract the key info $\hat{o}_s$ using the reasoner $\mathcal{R}$
5: $\mathcal{M}.\text{add}(\hat{o}_s)$  ▷ Update the working memory
6: **switch** $\hat{o}_s$ **do**
7:   **case** $\hat{o}_s$ is not informative
8:     $\text{goto}(2)$  ▷ Go to line 2 to make decision at the same state, excluding $t_s$.
9:   **case** $\hat{o}_s$ has useful information
10:     $state \leftarrow t_s$  ▷ Update state
11:     $\text{goto}(2)$  ▷ Go to line 2 to make decision for the next state.
12:   **case** $\hat{o}_s$ is ready as final answer
13:     $ans \leftarrow \hat{o}_s$  ▷ Output answer

---

Considering the potential intricacy of the task, we conduct a user study to gather a broad range of examples of human decision-making process, when using tools to respond to visual information-seeking queries (we introduce the details of data collection in Sec. 3.3). This helps us to establish a structured framework for decision-making. We utilize the data collected from this study to construct a transition graph $\mathcal{G}$ shown in Figure 3, which outlines all the possible actions at each given state. Additionally, we employ real-life decision-making examples $\mathcal{E}$, i.e., users choose which tool at different states, to guide the planner in choosing the appropriate action at each stage of the process.

The Algorithm 1 presents the operations of the planner $\mathcal{P}$. The planner undertakes a series of steps each time a decision is required regarding which tool to employ and what query to send to it. Firstly, based on the present $state$, the planner provides a range of potential subsequent actions $\mathcal{A}_s$. The potential action space $\mathcal{A}_s$ may be large, making the search space intractable. To address this issue, the planner refers to the human decisions from the transition graph $\mathcal{G}$ to eliminate irrelevant actions. The planner also excludes the actions that have already been taken before and are stored in the working memory $\mathcal{M}$. Formally, this procedure is $\mathcal{A}_s \leftarrow \phi(state, \mathcal{G}, \mathcal{M})$.

Next, it collects a set of relevant in-context examples $\mathcal{E}_s$ that are assembled from the decisions previously made by humans during the user study relevant to actions $\mathcal{A}_s$, that is $\mathcal{E}_s \leftarrow \theta(\mathcal{E}, \mathcal{A}_s)$. With the gathered in-context examples $\mathcal{E}_s$ and the working memory $\mathcal{M}$ that holds data collected from past tool interactions, the planner formulates a prompt, denoted by $p_s \leftarrow \psi(\mathcal{E}_s, \mathcal{M})$. The prompt $p_s$ is then sent to the LLM which returns a structured answer, determining the next tool $t_s$ to be activated and the query $q_s$ to be dispatched to it. We denote this action by $t_s, q_s \leftarrow LLM(p_s)$. This design allows the planner to be invoked multiple times throughout the process, thereby facilitating dynamic decision-making that gradually leads to answering the input query.

The Algorithm 2 shows the overall decision-making workflow of AVIS. The entire process repeats until a satisfactory answer is produced. Initially, the working memory is populated only with the input visual question $I$, and the initial $state$ is set to START. At each iteration, we first invoke the planner $\mathcal{P}$ to determine the next tool and the query to employ, as outlined in Algorithm 1. Subsequently, the selected external tool executes and delivers its output $o_s$. The output from the tools can be quite diverse, ranging from a list of identified objects, to a collection of similar images with their captions, to snippets of search results or knowledge graph entities.

Therefore, we employ a reasoner $\mathcal{R}$ to analyze the output $o_s$, extract the useful information and decide into which category the tool output falls: informative, uninformative, or final answer. Our method utilizes the LLM with appropriate prompting and in-context examples to perform the reasoning. If the reasoner concludes that it's ready to provide an answer, it will output the final response, thus concluding the task. If it determines that the tool output is uninformative, it will revert back to the

planner to select another action based on the current state. If it finds the tool output to be useful, it will modify the state and transfer control back to the planner to make a new decision at the new state.

Our approach, which employs dynamic decision-making coupled with backtracking, differs from previous methods [23, 36] that follow a plan-then-execute paradigm. Our system is structured to make decisions grounded to the results of current executions and to conduct iterative searches for tool combinations. This process eventually yields the most effective strategy to accomplish the task.

## 3.2 Tools and their APIs

To respond effectively to visual queries that necessitate in-depth information retrieval, it's important to equip AVIS with a comprehensive suite of tools. In this section, we describe these tools.

**Image Captioning Model**: We employ the PALI 17B [8] captioning model, which obtains state-of-the-art results for image captioning. This tool has the capability to generate captions for either the entire image or for a cropped image corresponding to the bounding box of a detected object.

**Visual Question Answering Model**: We utilize the PALI 17B [8] VQA model, which has been fine-tuned on the VQA-v2 [11] dataset. This tool takes an image and a question as inputs and provides a text-based answer as the output.

**Object Detection**: We use an object detector trained on a super-set of Open Images dataset [17] categories that is provided by Google Lens API [1]. We use high confidence threshold to only keep the top-ranked detected boxes for the input image.

**Image Search**: We utilize Google Image Search to obtain a broad range of information related to the image crop of a detected box as provided in Google Lens API [1]. This information encompasses various details, such as knowledge graph entities, titles of associated products, and captions of analogous or identical images. The availability of these details can vary based on the image crop input provided to Google Image Search. When it comes to decision-making, our planner considers the utilization of each piece of information as a separate action. This is due to the fact that each information could contain hundreds of tokens that necessitate complex processing and reasoning.

**OCR**: In some cases, images may include textual content such as street names or logos. To detect and utilize this text, we take advantage of the Optical Character Recognition (OCR) feature available in the Google Lens API [1].

**Web Search**: Web search enables our approach to acquire up-to-date world knowledge and retrieve relevant documents on any topic of interest. For this objective, we employ the Google Web Search API [2]. It accepts a text-based query as input and produces the following outputs: (i) related document links and snippets, (ii) in certain instances, a knowledge panel providing a direct answer to the query, and (iii) up to five questions that are related to the input query. If a knowledge panel is available, we parse it into a sentence or a few sentences that summarize its information.

**LLM short QA**: We incorporate a Language Model (LLM) powered question-answering component as another tool. This tool accepts a query in text form and produces an answer also in text form. It is important to note that the use of the LLM here as a question-answering tool is distinct from its role in the planner or reasoner as outlined in Alg. 1 and Alg. 2.

## 3.3 Gathering User Behavior to Inform LLM Decision Making

Many of the visual questions in datasets such as Infoseek [7], Oven [14] and OK-VQA [26] ask for fine-grained answers, which poses a challenge even for humans, often requiring the assistance of various APIs and web searches for answers. Figure 4(a) illustrates an example visual question taken from the OK-VQA [26] dataset. In order to gather insights into human decision-making process, we carried out a user study. More specifically, our goal is to understand how humans utilize external tools to answer visual queries that involve seeking information.

The user is equipped with an identical set of tools as our method. They are presented with the input image and question, along with image crops for each detected object. Additionally, tools like PALI Caption, PALI VQA, PALM, and Web Search are made available to the user. Furthermore, based on the information obtained through image search for each cropped image, the user is offered one or multiple buttons associated with each box. These buttons provide the user with the ability

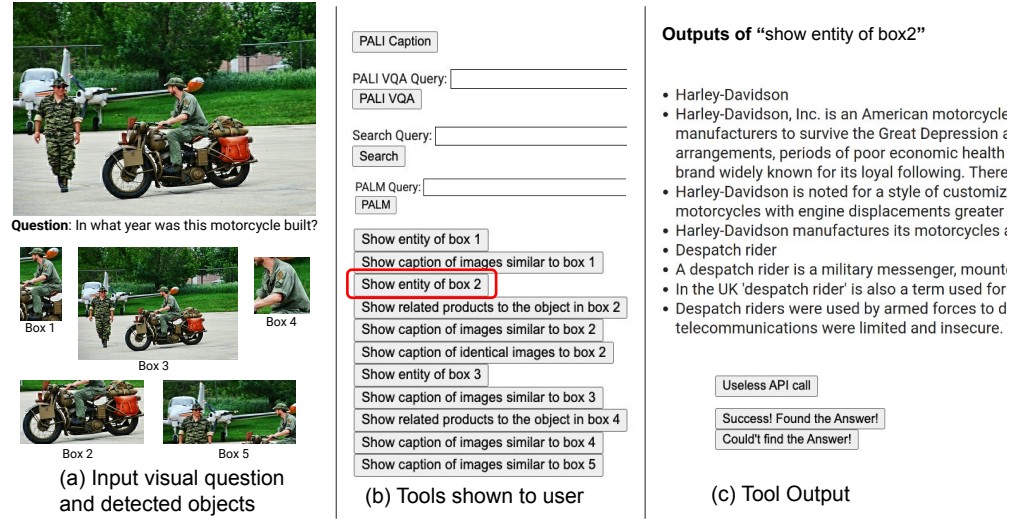

Figure 4: We conduct a user study to gather examples of user decision-making when responding to visual information-seeking questions. Given a visual question as depicted in (a), the user makes a series of tool calls using the available APIs shown in (b). Each tool call yields an output which the user reviews whether it is useful and determines the subsequent action, illustrated in (c).

to access diverse information pertaining to the image crop of the box. This includes details such as corresponding knowledge graph entities, captions of similar images, titles of associated related products, and captions of identical images. An example set of tools and APIs are shown in Figure 4(b).

When the user initiates an action, such as clicking on a button or submitting a query to web search, PALM, or PALI VQA, the corresponding tool is invoked, and the resulting output is displayed to the user. We record the sequence of actions taken by the user and the outputs that they receive at each step. For instance, in Figure 4, we show an example of how a user needs to perform four actions to answer the question: *i)* display entities in box 2, *ii)* show the caption of similar images to box 2, *iii)* conduct a search for *"In what year was Harley-Davidson XA built?"*, and *iv)* utilize PALM using the combination of the search output and the question *"In what year was Harley-Davidson XA built?"*. When the user is prepared to proceed to the next question, they click on either of the two buttons: "Success! Found the Answer!" or "Couldn't Find the Answer." Subsequently, a new visual question is presented to them.

The collected user behavior serves as a guide for our system in two key ways. Firstly, we construct a transition graph by analyzing the sequence of decisions made by users. This graph defines distinct states and restricts the available set of actions at each state. For example, at the START state, the system can take only one of these three actions: PALI caption, PALI VQA, or object detection. Figure 3 illustrates the transition graph that has been constructed based on the decision-making process of the users. Secondly, we utilize the examples of user decision-making to guide our planner and reasoner with relevant contextual instances. These in-context examples aid in enhancing the performance and effectiveness of our system.

We conducted a user study involving 10 participants who collectively answered a total of 644 visual questions. During the study, we presented users with visual questions that were randomly selected from both the Infoseek [7] and OK-VQA [26] datasets. This approach allowed us to provide the participants with a varied and diverse set of visual questions to assess and respond to. We show the details for this study as well as example prompts in the Appendix.

## 4 Experiments

We evaluate AVIS on two visual question answering datasets: *i)* OK-VQA [26], which requires common-sense knowledge not observed in given image; and *ii)* Infoseek$_{\text{wikidata}}$ [7], which further necessitates more fine-grained information that cannot be covered by common sense knowledge.

**Experimental Setup.** We follow the decision-making workflow in Alg. 2 to implement AVIS to solve visual questions. For the Planner, we write the basic instructions for describing each tool, and keep a pool of real user behavior when they select each tool, which we collected in the user study. At each

| Model | Unseen Entity | Unseen Question |
|---|---|---|
| PALM [9] (Q-only, few-shot) | 3.7 | 5,1 |
| OFA [22] (fine-tune) | 9.7 | 14.8 |
| PALI [6] (VQA, zero-shot) | 1.8 | 2.2 |
| PALI [6] (fine-tune) | 16.0 | 20.7 |
| PALM [9] w/ CLIP [32] (few-shot + external knowledge) | 21.9 | 18.6 |
| FiD [45] w/ CLIP [32] (fine-tune + external knowledge) | 20.7 | 18.1 |
| (—baselines without dynamic decision making, sequentially execute the tools—) | | |
| baseline-PALM w/ (PALI*, few-shot) | 12.8 | 14.9 |
| baseline-PALM w/ (PALI* + Object, few-shot) | 31.3 | 36.1 |
| baseline-PALM w/ (PALI* + Object + Search, few-shot) | 36.1 | 38.2 |
| **AVIS** (ours, few-shot) | **50.7** | **56.4** |
| w/o PALI* | 47.9 | 54.2 |
| w/o Object | 41.2 | 48.4 |
| w/o Search | 42.5 | 49.6 |

Table 1: **Visual Question Answering** results (accuracy) on Infoseek$_{Wikidata}$. The first four rows are results from their paper that do not use external knowledge, and the next two are from their paper that use CLIP as knowledge source. The tool PALI* denotes the frozen multi-task PALI-17B model for both visual question answering and image captioning. Object means object detection, and search means image and text search.

step $s$, we prepare the prompt based on the feasible action lists $\mathcal{A}_s$. For the Reasoner, we write the prompt for all APIs that return a long list of results, including *Object Detection*, *Product Detection*, *Web Image Search* and *Web Text Search*, that guides reasoner to extract the relevant information. Note that we design the reasoner in a way such that the "uninformative" answers can be detected. In order to support this, we manually prepare several bad examples that do not provide any useful information, pass it to the reasoner as a part of the prompt. We show the detailed prompts for these two modules in the Appendix.

We use the frozen PALM 540B language model [9] for both the planner and the reasoner, with deterministic generation ensured by setting the temperature parameter to zero. We use 10 examples as in-context prompts for each dataset, and report the VQA accuracy [11] as the evaluation metric.

**Baselines.** A significant novelty of AVIS is the ability to dynamically determine the relevant tools according to different states. To show that this design choice is useful, we add a number of baselines that do not contain a LLM-planner for dynamic decision making. Instead, they follow a pre-determined sequence to call a list of tools. We propose the following baselines:

- **baseline-PALM w/ PALI***, which integrates the captions generated by PALI and the visual answers from PALI VQA. PALI* denotes the combination of both VQA and captioning tool.
- **baseline-PALM w/ (PALI* + Object)**, which in addition calls the object detection tool, and then integrates all object data, including products and text detected by OCR.
- **baseline-PALM w/ (PALI* + Object + Search)**, a model which first selects a relevant object with the help of PALM, then sequentially executes the image search and Google search with the object name. It then calls PALM again to answer the question.

For each of the three baselines, we prepare a few-shot Chain-Of-Thought (COT) prompting [39], in which the COT prompt guides the model to explain why predictions are made based on the provided information. Note that these baselines utilize a set of tools in a fixed order, without the capacity for dynamic decision making.

We also evaluate the usefulness of each tool group (i.e., PALI*, Object, and Search) through an ablation study. This involves removing each tool group from our framework individually, and assessing the impact on performance.

**Experimental Results.** Table 5 presents the results of AVIS and other baselines on the Infoseek$_{wikidata}$ dataset. Infoseek$_{wikidata}$ is a challenging dataset that requires identifying highly specific entities. Even robust visual-language models, such as OFA [22] and PALI [6], fail to yield

| | Model | Accuracy (%) |
|---|---|---|
| **Supervised** | KRISP [25] | 38.4 |
| | KAT [12] | 54.4 |
| | ReVIVE [20] | 58.0 |
| | REVEAL [15] | 59.1 |
| | PALI [6] (OK-VQA, finetune) | 64.5 |
| **Zero-shot** | PALI [6] (VQA, zero-shot) | 41.6 |
| | PICa-Full [41] | 48.0 |
| | Flamingo (zero-shot) [4] | 50.6 |
| | BLIP-2 [18] | 45.9 |
| **Few-shot** | ViperGPT (one-shot) [36] | 51.9 |
| | Flamingo (few-shot) [4] | 57.8 |
| | (baselines without dynamic decision making, sequentially executing the tools) | |
| | baseline-PALM w/ (PALI*) | 44.3 |
| | baseline-PALM w/ (PALI*+Object) | 38.2 |
| | baseline-PALM w/ (PALI*+Object + Search) | 47.9 |
| | **AVIS** (ours) | **60.2** |
| | w/o PALI* | 47.1 |
| | w/o Object | 58.3 |
| | w/o Search | 55.0 |

Table 2: **Visual Question Answering** results (accuracy) on OK-VQA. The tool PALI* denotes the frozen multi-task PALI-17B model for both visual question answering and image captioning. Object means object detection, and search means image and text search.

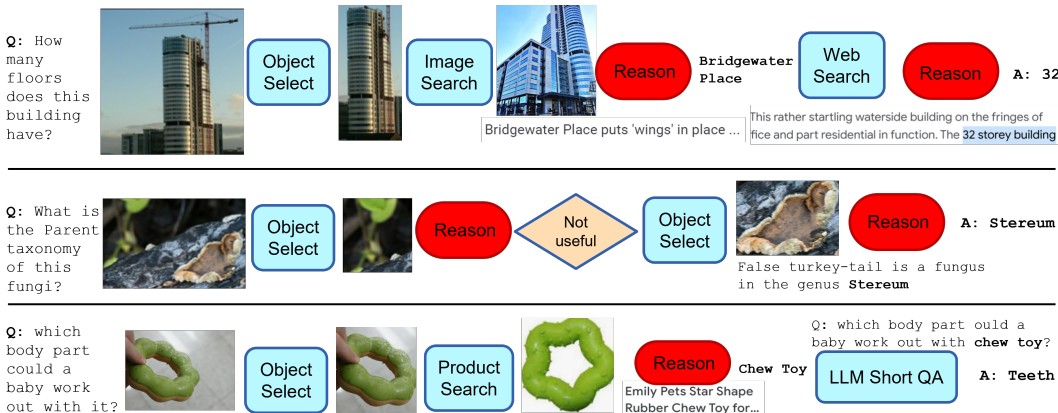

Figure 5: Examples of AVIS's dynamic planning and reasoning procedure for solving visual questions.

high accuracy when fine-tuned on this dataset. However, our AVIS, without fine-tuning and by leveraging a complete set of tools guided by 10 in-context examples, achieves the accuracy of 50.7 and 56.4 on the unseen entity and question splits, respectively. This significantly outperforms the fine-tuned results of PALI-17B, which are 16.0 and 20.7, as well as the PALM model augmented with CLIP knowledge, which are 21.9 and 18.6, respectively.

Table 5 also illustrates that our improvements are not solely due to the additional information provided by the external tools, but due to our dynamic decision-making pipeline. We compare the results of AVIS with the three baselines that conduct sequential execution. While these baselines do improve the performance, our AVIS framework outperforms the best baseline model by up to 17.3 accuracy. Note that AVIS and the baselines use exactly the same set of tools. This considerable performance gap clearly shows the clear advantage of our dynamic decision-making design. Furthermore, we show the importance of each tool in the last block of Table 5. Removal of any of the tools degrades the overall accuracy. Among the three tool groups, Object and Search are more important than PALI, as they provide more fine-grained information crucial for the Infoseek dataset.

We report the OK-VQA experiments in Table 2. AVIS with few-shot in-context examples achieves an accuracy of 60.2, higher than most of the existing methods tailored for this dataset, including KAT [12], ReVIVE [20] and REVEAL [15] . AVIS achieves lower but comparable performance compared to PALI model fine-tuned on OK-VQA. This difference, compared to Infoseek, may be attributed to the fact that most QA examples in OK-VQA rely more on commonsense knowledge than on fine-grained knowledge. Therefore, it is feasible to encode such generic knowledge in the model parameters and requires less external knowledge. Note that PALI zero-shot VQA model itself achieves 41.6 accuracy, which is significantly higher than in Infoseek, which supports this hypothesis. Table 2 also shows that the object detection is less crucial as a tool on this data set, compared to PALI captioning and VQA. Nonetheless, AVIS equipped with all tools achieves the best performance.

**Case studies for dynamic decision making.** One of the key features of AVIS is its ability to dynamically make decisions instead of executing a fixed sequence. Figure 5 presents three examples of AVIS's dynamic planning and reasoning process. They demonstrate the flexibility of AVIS to use different tools at various stages. It is also worth noting that our reasoner design enables AVIS to identify irrelevant information, backtrack to a previous state, and repeat the search. For instance, in the second example concerning the taxonomy of fungi, AVIS initially makes an incorrect decision by selecting a leaf object. However, the reasoner identifies that this is not relevant to the question, prompting AVIS to plan again. This time, it successfully selects the object related to false turkey-tail fungi, leading to the correct answer, Stereum. Some detailed error analysis is shown in Appendix F.

## 5    Conclusion

In this paper, we propose a novel approach that equips the Large Language Models (LLM) with the tree-search to use a variety of tools for answering knowledge-intensive visual questions. Our methodology, anchored in human decision-making data collected from a user study, employs a structured framework that uses an LLM-powered planner to dynamically decide on tool selection and query formation. An LLM-powered reasoner is tasked with processing and extracting key information from the output of the selected tool. Our method iteratively employs the planner and reasoner to leverage different tools until all necessary information required to answer the visual question is amassed.

**Limitation Statement:** Currently AVIS is specifically designed for visual question answering. We aim to extend our LLM-powered dynamic decision-making framework to address other reasoning tasks. Additionally, our current framework depends on a computationally intensive LLM, namely, the PALM model. We are interested in investigating whether this decision-making framework can also be performed by lighter weight language models.

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
