## A    Implementation of AVIS workflow

We implemented AVIS using the code snippet referenced in Code 7. Throughout our experiments, we employed the APIs of Google Search, LENS, PALI, and PALM directly, without the need for additional GPU/TPU computational resources. Tools that didn't require input queries, such as object detection, captioning, and image search, had their results pre-calculated over the two datasets to reduce the time cost. Other services like VQA, text search, and LLM QA were called during runtime.

## B    Comparison to pure Autonomous baseline without Transition Graph

One of the significant contributions of this paper lies in the use of a transition graph, synthesized from an authentic user study. To underscore the importance of this graph, along with user prompts in facilitating the efficacy of AVIS, we devised a baseline that operates independently of the transition graph. In this scenario, the model, at each timestep, is presented with a comprehensive list of all tools, each paired with a task description. This baseline shares similarities with the recently launched AutoGPT [2], BabyAGI[3] projects, which attempted to utilize LLMs as autonomous agents to select all possible actions available in the web.

The results are show in Table 3 on Infoseek WIkiData unseen entity set and OKVQA. Note that this baseline doesn't achieve the number as high as AVIS with the transition graph and user prompts. The key reason for this discrepancy is the global characteristics inherent in the tool list we have. For instance, we typically first address the visual sub-question through object detection and image search, followed by resolving the knowledge component via Google Search and LLM. However, solely relying on the task description, devoid of human behavior as guidance, can result in the model generating unrealistic tools. We will discuss this intuition more in the following sections.

| Model | Infoseek | OKVQA |
|---|---|---|
| AVIS w.o/ Transition Graph | 38.2 | 47.3 |
| AVIS w/ Transition Graph | 50.7 | 60.2 |

Table 3: Ablation of AVIS with or without the guidance of Transition Graph

## C    Analysis of AVIS's generated tool execution sequence

We have also conducted an analysis to determine whether common patterns exist within the generated programs of AVIS's predictions.

We gathered the tool execution traces for all samples within the Infoseek unseen entity dataset. Initially, we display the frequency of each tool being invoked in Figure 6, followed by a more detailed analysis of the first to fourth most commonly called tools in Figures 7-10. As illustrated, the AVIS model, guided by the transition graph and prompts, does not utilize all possible combination of tools, but favors some certain combinations. For instance, as depicted in Fig 7, "object select" is utilized more frequently than other tools at the outset. Similarly, as demonstrated in Fig 9, during the third step, when the model accumulates the visual answer, it is likely to invoke "web search" to gather additional information.

We have also calculated the transition probability of the induced graph in Fig 11. The structure of this graph differs slightly from the guided transition graph because during actual runtime, the model will not predict some of the edges. Overall, it reveals a clear two-step question-solving pattern. Initially, AVIS gathers sufficient visual information through the use of visual tools such as "object detection," "VQA," or "identical image search," and then employs "LLM QA" to obtain the visual answer. Subsequently, it iteratively calls "web search" and "LLM QA" post-search with a prompt, eventually deriving the final answer. We also present the distribution of the lengths of generated sequences in Figure 13. As illustrated, the lengths vary considerably, rather than maintaining a fixed value, with a length of 5 being most common for the generated sequences.

---

[2]https://github.com/Significant-Gravitas/Auto-GPT
[3]https://github.com/yoheinakajima/babyagi

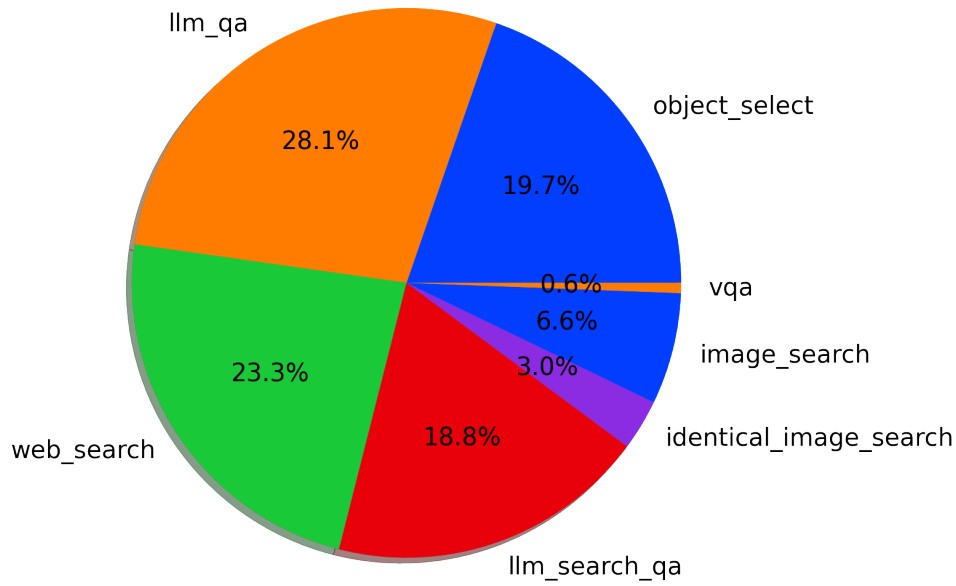

Figure 6: Overall frequency of tool usage on Infoseek dataset.

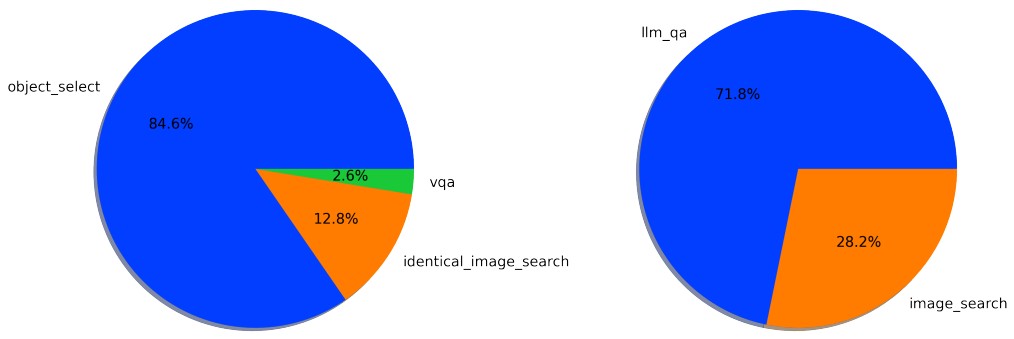

Figure 7: Frequency of the first used tool.   Figure 8: Frequency of the second used tool.

Another intriguing aspect worth exploring is our reasoner component. As explained in the paper, the reasoner evaluates whether the output of each tool is "informative," "not informative," or "answerable". We exhibit the overall frequency of these predictions in Figure 12. As shown, the model tends to classify most of the outputs as either informative or answerable. However, approximately 8.1% of returned entries are deemed "not informative," in which case AVIS would backtrack to select alternative actions. We further demonstrate a few examples of different choices in Table 4.

## D   Dataset Details

**Infoseek**[4] is a Visual Question Answering (VQA) dataset, specifically geared towards information-seeking questions that cannot be answered merely through common sense knowledge. This dataset was curated by initially gathering human-annotated questions, which were then automatically integrated with existing visual entity recognition datasets and Wikidata to generate complex question-

---

[4]https://open-vision-language.github.io/infoseek/

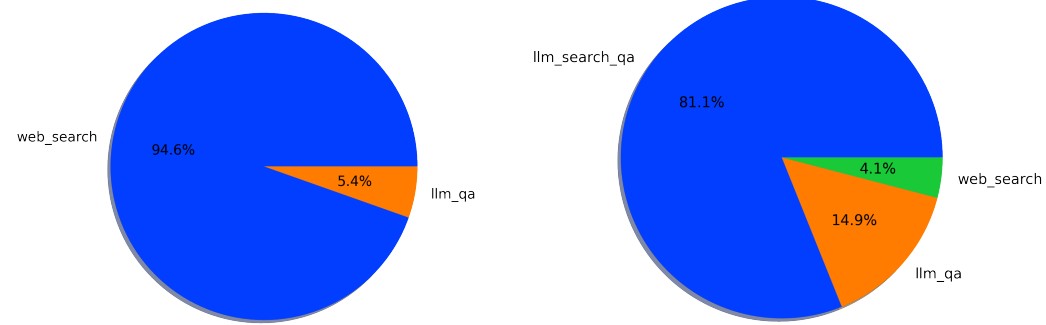

Figure 9: Frequency of the third used tool.     Figure 10: Frequency of the forth used tool.

| Question | Action | Returned Content | Output of Reasoner |
|---|---|---|---|
| where is the island located in? | image search | Retrievals: [willow - Students \| Britannica Kids \| Homework Help (score=57.2), Silky Willow Live Stakes For Sale \| Buy Live Stakes Wholesale (score=55.3), ...] | not informative |
| Who or what maintains this place? | LLM QA | the history of trafalgar square. the most relevant knowledge entry about trafalgar square is "Trafalgar Square is the focal point for protests, revelry and victory celebrations, however it was only in 1835 that the area became known by this name". From the result, we know the history of trafalgar square is trafalgar square is the focal point for protests, revelry and victory celebrations, however it was only in 1835 that the area became known by this name. Terefore, the predicted answer is trafalgar square is the focal point for protests , revelry and victory celebrations , however it was only in 1835 that the area became known by this name. | not informative |
| What is the total quantity of produced items for this type of aircraft? | image search | Retrievals: [high confidence: Condor bietet neue Langstrecke nach Brasilien an - AERO International (score=97.3), high confidence: Air safety incidents for aircraft registration D-ABOE - AeroInside (score=95.0), ...] | yes, answerable |
| what is the name of this mountain? | image search | Object: [ a view of a castle in the distance . There are trees and buildings present at the bottom of this image. We can see a hill and a tower in the middle of this image. We can see the sky in the background. (Caption, whole image), Mount of Olives (ridge): The Mount of Olives or Mount Olivet is a mountain ridge east of and adjacent to Jerusalem's Old City (score=88.6), Mount Zion (peak): Mount Zion is a hill in Jerusalem, located just outside the walls of the Old City (score=79.0)] | yes, informative |

Table 4: Several examples of API execution results and the reasoner's justification.

answer pairs. At the time of submission, we only have access to its wikidata split. Here we also report the results on human split in Table 5.

**OK-VQA**[5] is another VQA dataset, unique in its requirement for the application of external knowledge that transcends the information directly visible in the input images. The creation of this dataset involved crowdsourced workers who were tasked with annotating complex questions, drawing upon the extensive knowledge resources available on Wikipedia.

# E    Prompt Examples

Below we show different prompt examples to support our AVIS workflow. First is the prompts for planning, which selects which tool to use and what query to send. It is consists of a overall task descriptions and many real examples showing at which circumstances real users select this tool.

```
1 planner_prompt =
2 """You goal is to answer the following query: %s.
3
4 To answer it, you will be provided with the following tools:
5 %s
6
7 Please make the decision based on the current context.
8
9 %s
```

---
[5] https://okvqa.allenai.org/

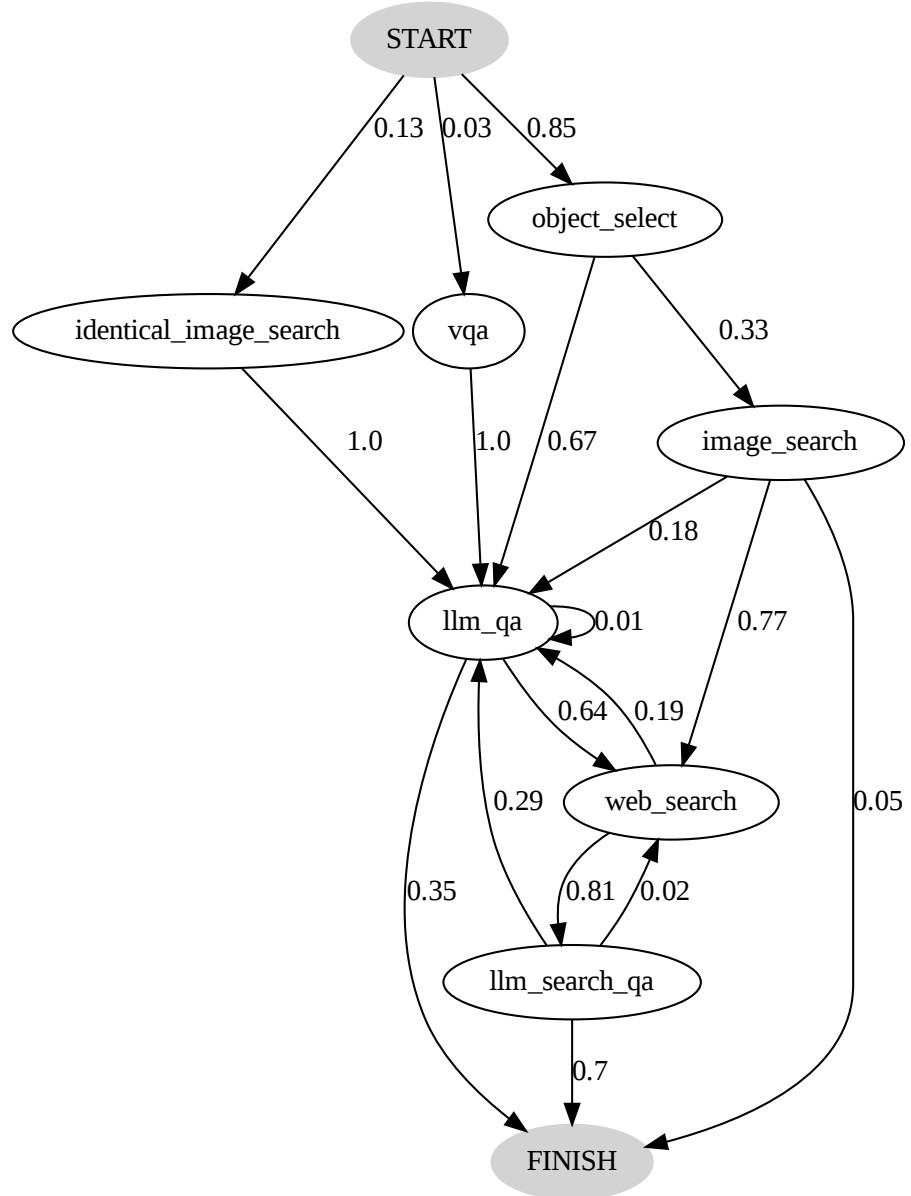

Figure 11: Induced transition frequency graph of AVIS over Infoseek dataset.

```
10 Query: %s
11 Context: %s
12 Action: \n
13 """
14
15 task_instructions = {
16 'vqa':
17     'You will ask simple question about this image to a external QA module. Please use this when the input
        query is very straightforward and simple.',\
18 'object_select':
19     'You will select one of the object we detect to dig further. Please use when the question asks about a
        specific object.',\
20 'identical_image_search':
```

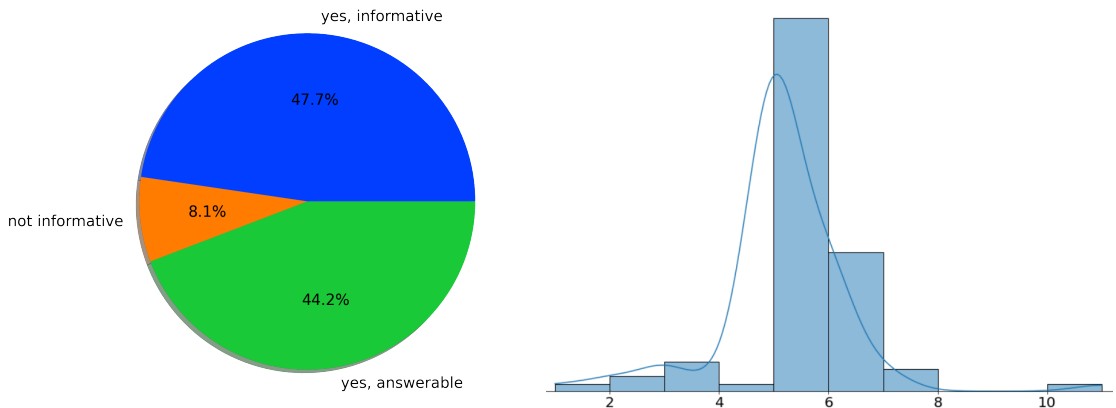

Figure 12: Overall frequency of judgement by reasoner of AVIS.

Figure 13: Length distribution of AVIS's generated action sequences.

| Model | Unseen Entity | Unseen Question |
|---|---|---|
| PALM (Q-only, few-shot) | 6.6 | 4.8 |
| OFA (fine-tune) | 2.9 | 6.2 |
| PALI (fine-tune) | 5.9 | 13.3 |
| PALM w/ CLIP (few-shot + external knowledge) | 14.9 | 15.6 |
| FiD w/ CLIP (fine-tune + external knowledge) | 17.6 | 18.9 |
| **AVIS** (ours, few-shot) | **31.4** | **33.6** |

Table 5: **Visual Question Answering** results (accuracy) on Infoseek$_{human}$. The first four rows are results from their paper that do not use external knowledge, and the next two are from their paper that use CLIP as knowledge source.

```
21    'You will see captions of all images identical to the given image. Please use when the question asks
        about the whole image instead of a part.',\
22  'image_search':
23    'You will see captions of all images similar to this object. Please use when you need more information.',\
24  'web_search':
25    'You will send question to Google Search to get knowledge. Please use when the current query requires
        extra knowledge',\
26  'llm_qa':
27    'You will send question to a QA module. Please use this when the input query is simple and contain
        common-sense knowledge'
28 }
```

Listing 1: Planner prompt skeleton and Task instructions

```python
vqa_plan_prompts = [
"""Query: what is the train carrying?
Context: [
  a train traveling down train tracks next to a forest . There are four trains on the railway track. In the
      background there are trees,poles and sky. (Caption, whole image)
  Extracted Text: BNSF (score=100.0),
  BNSF Railway: BNSF Railway is one of the largest freight railroads in North America (score=89.3),
]
Action: vqa
""",\
"""Query: What is the girl wearing on her legs?
Context: [
  a woman standing in a field putting on a coat . There is a woman standing on the ground. This is grass and
      there are plants. In the background we can see some trees and this is sky. (Caption, whole image)
]
Action: vqa
""",\
"""Query: what color is the bus?
Context: [
  a double decker bus parked in front of a building . There is a double decker bus on the road and this is
      snow. Here we can see a pole, light, trees, and houses. In the background there is sky. (Caption, whole
      image)
  Extracted Text: ENVIRO400 (score=100.0),
  Extracted Text: Les Miserables (score=100.0),
  Query Suggestion: les miserables (score=100.0),
  Volvo Olympian: The Volvo Olympian was a rear-engined 2-axle and 3-axle double decker bus chassis
      manufactured by Volvo at its Irvine, Scotland factory (score=88.5),
  Alexander Dennis Enviro400: The Alexander Dennis Enviro400 is a twin-axle low-floor double-decker bus that
      was built by the British bus manufacturer Alexander Dennis between 2005 and 2018 (score=85.4),
]
Action: vqa
""",\
"""Query: what is the person doing?
Context: [
  two people sitting on the floor opening presents . There are sofas on the sofas there are pillows, here
      there is table, on the table there are plants and other objects, here there are two persons sitting on
      the ground, gift boxes, dog and this is floor. (Caption, whole image)
]
Action: vqa
"""
]
object_select_plan_prompts = [
"""Query: what is the name of this building?
Context: [
  a group of people that are standing in front of a building . There is a building in the left corner which
      has few people standing in front of it and there is a fire hydrant in the right corner and there is a
      street light pole beside it. (Caption, whole image)
  Query Suggestion: Alcatraz Warden's House San Francisco (score=100.0),
  Alcatraz Island (historic_site): Alcatraz Island is a small island 1 (score=91.9),
  Warden's House: The Warden's House was the home of the wardens of the federal penitentiary on Alcatraz
      Island, off San Francisco (score=78.1),
]
Action: object_select
""",
"""Query: what is the island?
Context: [
  a view of a mountain from a cable car . There is a ropeway. Behind that there are trees and hills.
      (Caption, whole image)
  Ngong Ping 360 (gondola_lift_station): Ngong Ping 360 is a bicable gondola lift on Lantau Island in Hong
      Kong (score=91.8),
  Tian Tan Buddha (monument): The Big Buddha is a large bronze statue of Buddha, completed in 1993, and
      located at Ngong Ping, Lantau Island, in Hong Kong (score=79.0),
]
Action: object_select
""",
"""Query: what is the name of this place?
Context: [
  a cemetery with a building in the background . There is a road and there are many atoms and trees beside it
      and there is a building in the right corner. (Caption, whole image)
]
Action: object_select
""",
"""Query: what is the name of this bird?
Context: [
  a bird sitting on top of a lush green hillside . There is a bird on the grassland in the foreground area of
      the image and the background is blurry. (Caption, whole image)
  Atlantic puffin (type_of_bird): The Atlantic puffin, also known as the common puffin, is a species of
      seabird in the auk family (score=73.2),
  Horned puffin (type_of_bird): The horned puffin is an auk found in the North Pacific Ocean, including the
      coasts of Alaska, Siberia and British Columbia (score=73.2),
  Puffins (type_of_bird): Puffins are any of three species of small alcids in the bird genus Fratercula
      (score=73.2),
  Fraterculini (score=48.8),
  Auk (type_of_bird): An auk or alcid is a bird of the family Alcidae in the order Charadriiformes
      (score=11.8),
]
Action: object_select
"""
]
identical_image_search_plan_prompts = [
"""Query: what is the name of this building?
Context: [
```

```
73    a group of people that are standing in front of a building . There is a building in the left corner which
          has few people standing in front of it and there is a fire hydrant in the right corner and there is a
          street light pole beside it. (Caption, whole image)
74    Query Suggestion: Alcatraz Warden's House San Francisco (score=100.0),
75    Alcatraz Island (historic_site): Alcatraz Island is a small island 1 (score=91.9),
76    Warden's House: The Warden's House was the home of the wardens of the federal penitentiary on Alcatraz
          Island, off San Francisco (score=78.1),
77  ]
78  Action: identical_image_search
79  """,
80  """Query: what is the aircraft?
81  Context: [
82    a fighter jet sitting on top of an airport tarmac . There is a plane and missiles on the ground. At the
          left a person is standing wearing a cap. (Caption, whole image)
83    Extracted Text: AIRLINERS.NET (score=100.0),
84    Query Suggestion: airliners.net (score=100.0),
85    Airliners: Airliners (score=74.8),
86    British Aerospace Hawk 200: The British Aerospace Hawk 200 is a single-seat, single engine light multirole
          fighter designed for air defence, air denial, anti-shipping, interdiction, close air support, and
          ground attack (score=74.8),
87    product: Airfix BAE Hawk T1 1:72 (score=0.0),
88    product: Rolls-royce Adour In The Hawk / Bae Hawk 200 . Pdf/download (score=0.0),
89  ]
90  Action: identical_image_search
91  """,
92  """Query: what is the name of this place?
93  Context: [
94    a row of pillars sitting next to a dirt road . There is a building and this is plant. Here we can see
          pillars and a sky. (Caption, whole image)
95    Query Suggestion: Palmyra Archaeology (score=100.0),
96    Great Colonnade at Palmyra (ancient_roman_architecture_structure): The Great Colonnade at Palmyra was the
          main colonnaded avenue in the ancient city of Palmyra in the Syrian Desert (score=90.3),
97  ]
98  Action: identical_image_search
99  """,
100 """Query: what is the name of this lake?
101 Context: [
102   a view of a river surrounded by mountains . There are trees in the right corner and there is a river and
          mountains in front of it. (Caption, whole image)
103   Monte Bre (peak): Monte Bre is a small mountain east of Lugano on the flank of Monte Boglia with a view of
          the bay of Lugano and the Pennine Alps and the Bernese Alps (score=85.5),
104   product: Top Searched (score=0.0),
105 ]
106 Action: identical_image_search
107 """
108 ]
109 action_prompt_dict = {'vqa': vqa_plan_prompts, 'object_select': object_select_plan_prompts,
          'identical_image_search': identical_image_search_plan_prompts, 'image_search':
          image_search_plan_prompts, 'web_search': web_search_plan_prompts,
110 'llm_qa': llm_qa_plan_prompts}
```

Listing 2: Planning Prompts Example

We then show how AVIS decompose question into a visual sub-question and a knowledge sub-question. This is done at beginning to guide later tool usage.

```
1  question_decomposition_prompt = """
2      Read the following question for a given image. Decompose the question into two sub-questions.
3
4      The first will ask information about the image, and the second requires reasoning over the textual
         knowledge.
5      In the second question, we use # to denote the answer of the first question.
6
7
8      Question: what chemical makes the vegetable orange?
9      Visual: which orange vegetable is shown?
10     Knowledge: chemical makes # orange?
11
12
13     Question: How long can their horns grow?
14     Visual: which animals are shown?
15     Knowledge: How long can #'s horns grow?
16
17
18     Question: What is a competition for these animals called?
19     Visual: which animals are shown?
20     Knowledge: competition for #?
21
22
23     Question: What is the name of the ancient greek sport that evolved into the sport featured above?
24     Visual: which sport is played?
25     Knowledge: name of the ancient greek sport that evolved into #?
26
27
28     Question: Which food item here has the most protein?
29     Visual: what are the food items shown?
30     Knowledge: Which food item of # has the most protein?
31
32
33     Question: How many calories are in this meal?
```

```
34    Visual: what are the food items shown?
35    Knowledge: calories in #?
36
37
38    Question: What type of sandwich is this?
39    Visual: which type of sandwich is shown?
40    Knowledge: #
41
42    Question: What is the name of the restaurant where this was served?
43    Visual: which food items are served?
44    Knowledge: restaurant where # was served?
45
46
47    Question: What genus of bird is flying here?
48    Visual: what genus of bird is flying?
49    Knowledge: #
50
51
52    Question: What is the main ingredient in this food?
53    Visual: which food is shown?
54    Knowledge: main ingredient in #?
55 """
```

Listing 3: Question Decomposition Prompts

Below are several examples to help AVIS learns how to select the most suitable object ID.

```
 1 object_select_prompt = """
 2    Please think step by step. In the following, you will be given a "Query", a list of "Objects".
 3
 4    Your task is to predict the object #ID that is mostly relevant to answer the querys. Please generate the
        detailed explanation why you select this object, and then output ID in "Object #ID".
 5
 6
 7 Query: which city is this place?
 8 Object #0 [
 9  a row of pillars sitting next to a dirt road . There is a building and this is plant. Here we can see
        pillars and a sky. (Caption, whole image)
10  Query Suggestion: Palmyra Archaeology (score=100.0),
11  Great Colonnade at Palmyra (ancient_roman_architecture_structure): The Great Colonnade at Palmyra was the
        main colonnaded avenue in the ancient city of Palmyra in the Syrian Desert (score=90.3),
12 ]
13 Object #1 [
14  a green plant sitting next to a brick wall . There is a plant and this is wall. And there is a sky.
        (Caption, center)
15  Date palm (type_of_palm_trees): Phoenix dactylifera, commonly known as date palm, is a flowering plant
        species in the palm family, Arecaceae, cultivated for its edible sweet fruit called dates (score=81.7),
16 ]
17 Object #2 [
18  a wicker basket sitting on top of a rock . There is a blur image of a rock. (Caption, lower right)
19 ]
20 Output: The query asks about the city of the place. Only Object #0 (whole image) mentions city name Palmyra,
        which is an acient city. Also, Object #0 contains Query Suggestion "Palmyra Archaeology".
21 Therefore, the predicted Object #ID is 0.
22
23
24 Query: where is this place?
25 Object #0 [
26  a view of a valley surrounded by mountains . There are hills and this is grass. Here we can see trees and
        this is sky. (Caption, whole image)
27 ]
28 Object #1 [
29  a view of a lush green hillside with trees . There is a house on the rock and there are few plants beside
        it and there is a greenery ground in the background. (Caption, center)
30  Monterey Pine (type_of_conifers): Pinus radiata, the Monterey pine, insignis pine or radiata pine, is a
        species of pine native to the Central Coast of California and Mexico (score=49.1),
31  European rabbit (type_of_leporids): The European rabbit or coney is a species of rabbit native to the
        Iberian Peninsula, western France, and the northern Atlas Mountains in northwest Africa (score=31.3),
32 ]
33 Object #2 [
34  a green plant growing on a rocky surface . There is a blur image of trees and rocks. (Caption, lower center)
35  product: GreenView Fairway Formula Seed Success Paillis biodegradable avec engrais Sac de 4,5 kg Couvre 200
        m2 (score=0.0),
36 ]
37 Object #3 [
38  a rocky hillside with lots of green vegetation . There are trees and this is rock. (Caption, lower left)
39  Willow: Willows, also called sallows and osiers, of the genus Salix, comprise around 350 species of
        typically deciduous trees and shrubs, found primarily on moist soils in cold and temperate regions
        (score=31.3),
40  Tamarisk: The genus Tamarix is composed of about 50-60 species of flowering plants in the family
        Tamaricaceae, native to drier areas of Eurasia and Africa (score=26.8),
41 ]
42 Output: The query asks about the location of this place. Although these entries doesn't explicitly contain
        location name, but Object #1 (center) contains Monterey Pine and European rabbit, which might hint the
        location later.
43 Therefore, the predicted Object #ID is 1.
44 """
```

Listing 4: Object Select Prompts

Below are the prompts to extract answer from objects and extracted captions of similar images.

```
1 reason_vqa_prompt = """
2 Please think step by step. In the following, you will be given:
3
4 - Query: The query to be asked.
5 - Think: Why the following knowledge is retrieved.
6 - Entity: A list of entities that describe the object.
7 - Retrievals: A list of web documents that are similar to the object. If there's "high confidence", it's very
      important.
8
9 Your task is to predict a short answer to the query based on the provided information. You need to first
      identify which knowledge entry is mostly relevant, and then extract the answer from the knowledge.
10 Rely on Object information more, and if there contains "Query Suggestion", try to use it. Otherwise, if a
      information appears lots of time, there's a higher chance it's the answer.
11 After explaining your decision choice, saying "Answer is" and appending your predicted short answer. Please
      also generate the type of the answer after a comma.
12 If you are uncertain about the answer, especially when the knowledge is irrelevant to the query, say "cannot
      be answered". Do not generate the answer not inside the provided knowledge.
13
14
15
16 Query: what is this building?
17 Think: object  (whole image) contains stockholm city hall, which is the seat of stockholm municipality in
      stockholm, sweden.
18 Object: [
19   Stockholm City Hall (city_hall): Stockholm City Hall is the seat of Stockholm Municipality in Stockholm,
        Sweden (score=96.1),
20   Bla Hallen (banquet_hall): The Blue Hall is the main hall of the Stockholm City Hall best known as the
        banquet hall for the annual Nobel Banquet, and also used for state visits, student balls, jubilees and
        other large events (score=79.0),
21 ]
22 Retrievals: [
23   high confidence: City Hall - Blue Hall (1) | Stockholm (2) | Pictures | Sweden in Global-Geography
        (score=47.8),
24   high confidence: le salon bleu a city hall (salle de remise des prix nobel) - Picture of Stockholm,
        Stockholm County - Tripadvisor (score=47.7),
25 ]
26
27 Output: The query asks about the building. From both Object and Retrievals, there are mentions about
      Stockholm City Hall and Blue Hall. As Stockholm City Hall contains Blue Hall, the answer shall be
      Stockholm City Hall.
28 Therefore, the predicted answer is Stockholm City Hall.
29
30
31 Query: which sport is played?
32 Think: Object shows a snail sitting on top of a tennis ball.
33 Object: [
34   Cantareus apertus (type_of_gastropods): Cantareus apertus, commonly known as the green garden snail, is a
        species of air-breathing land snail, a terrestrial pulmonate gastropod mollusc in the family Helicidae,
        the typical snails,
35   Garden snail (type_of_gastropods): Cornu aspersum, known by the common name garden snail, is a species of
        land snail in the family Helicidae, which includes some of the most familiar land snails,
36   Helix aspersa aspersa (type_of_gastropods),
37   Slug: Slug, or land slug, is a common name for any apparently shell-less terrestrial gastropod mollusc,
38   Snail: A snail is a shelled gastropod,
39 ]
40 Retrievals: [
41   2019 NEWBIE Competition Winner Steven Ryan, Snail Farming - YouTube,
42   Alive specimens. a. Megalobulimus ovatus (CMIOC 11136), b. Thaumastus... | Download Scientific Diagram,
43   Brown garden snail > Manaaki Whenua,
44   Common garden snail and baby,
45   Easy Everyday Food for Garden Snails - Ask the plantician,
46   Green Life Soil: Natural pest & disease control in a winter garden,
47   Helminthoglyptinae - Wikipedia,
48   Hydrosalpingitis in broilers - Veterinaria Digital,
49   Master Gardener: Protecting squash and cucumbers from slugs and snails - Press Enterprise,
50   Mother Baby Blue Snails On Phalaenopsis Stock Photo 530400856 | Shutterstock,
51 ]
52
53 Output: The query asks about sport. From both entities and retrievals, they only talk about snail, and there
      is no information about which sport is played.
54 Therefore, given the provided information, this query cannot be answered.
55
56
57 Query: which sport is played?
58 Think: object , object , and object all contain people playing basketball. however, object is the only one
      that contains a group of women playing basketball.
59 therefore, the predicted object #id is 0.
60 Retrievals: [
61   08.07.2011 Zanele Mdodana of South Africa in action during the Quarter-finals between New Zealand and South
        Africa, Mission Foods World Netball Championships 2011 from the Singapore Indoor Stadium in Singapore
        Stock Photo - Alamy,
62   55 Brazilian Handball Team Images, Stock Photos & Vectors | Shutterstock,
63   :::Malawi High Commission:::,
64   Amanda Mynhardt Photostream | Netball, Netball singapore, Netball south africa,
65   Australia pass Malawi test with flying colours at Netball World Cup | Netball World Cup 2019 | The Guardian,
66   Australia's Jo Weston (second left) and Barbados' Latonia Blackman in action during the Netball World Cup
        match at the M&S Bank Arena, Liverpool Stock Photo - Alamy,
67   Birmingham 29795 World Netball Championships Final Editorial Stock Photo - Stock Image | Shutterstock,
68   Bridget kumwenda malawi netball hi-res stock photography and images - Alamy,
```

```
69   England V Australia International Netball Series Photos and Premium High Res Pictures | Netball, Netball
         quotes, Inspirational women,
70   File:Xx0992 - Madrid basketball Donna Burns - 3b - Scan.jpg - Wikimedia Commons,
71 ]
72
73 Output: The query asks about which sport is played. From retrievals, there exist many mentions about netball,
         and mentions that they are played by women.
74 therefore, the predicted answer is women netball.
75
76
77
78 Query: what is the name of the insect?
79 Think: only object  (while image) mentions the name of the insect, western tiger swallowtail.
80 Object: [
81   Query Suggestion: Western Tiger Swallowtail (score=100.0),
82   Canadian tiger swallowtail (type_of_lepidoptera): Papilio canadensis, the Canadian tiger swallowtail, is a
         species of butterfly in the family Papilionidae (score=78.4),
83   Eastern tiger swallowtail (us_state_butterfly): Papilio glaucus, the eastern tiger swallowtail, is a
         species of butterfly native to eastern North America (score=78.4),
84 ]
85 Retrievals: [
86   high confidence: kupu-kupu - Wiktionary (score=100.0),
87   high confidence: Top Spots for Nature Watching and Birding | VisitMaryland.org (score=100.0),
88   high confidence: File:Eastern Tiger Swallowtail Papilio glaucus on Milkweed 2800px.jpg - Wikimedia Commons
         (score=99.8),
89   high confidence: Photographing Butterflies - Life in the Finger Lakes (score=97.8),
90 ]
91
92 Output: The query asks about the name of the insect. From Object, it contains a very informative "Query
         Suggestion: Western Tiger Swallowtail".
93 Therefore, the predicted answer is Western Tiger Swallowtail.
94
95 """
```

Listing 5: Reason Prompt (Visual Question)

Below are prompts AVIS extract answer from search results:

```
1 reason_qa_prompt = """
2    Please think step by step. In the following, you will be given a "Query", and a list of "Knowledge" from
        Google Search related to this query.
3
4    Your task is to predict a short answer to the query based on the provided information. You need to first
        identify the most relevant knowledge entry, and then predict a short answer based on the knowledge. If
        a information appears lots of time, there's a higher chance it's the answer.
5
6    After explaining your decision choice, saying "Answer is" and appending your predicted answer.
7    If you are uncertain about the answer, especially when the knowledge is irrelevant to the query, say
        "cannot be answered". Do not generate the answer not inside the provided knowledge.
8
9
10 Query: What chemical makes carrot orange?
11 Knowledge: [
12 Title: How did carrots become orange? - The Economist
13 Content: High Confidence Response: carotenoids.
14
15 Context: The chemical compounds that give carrots their vivid colour, carotenoids, are usually used by plants
        that grow above ground to assist in the process of photosynthesis.
16
17 Title:
18 Content: carotenoids
19
20 The chemical compounds that give carrots their vivid colour, carotenoids, are usually used by plants that
        grow above ground to assist in the process of photosynthesis.
21
22 Title: Can Eating Too Many Carrots Make Your Skin Turn Orange? | Britannica - Encyclopedia Britannica
23 Content: Maybe not! Carrots and other orange fruits and vegetables are rich in a pigment known as
        beta-carotene. In humans, this pigment is converted to vitamin A by specialized cells in the small
        intestine. When high levels of beta-carotene are consumed, not all of the pigment is converted to
        vitamin A.
24 Fortunately, the skin discoloration fades when the diet is changed and the levels of beta-carotene in the
        blood decline.
25
26 Title: Why are carrots orange? | Ask Dr. Universe | Washington State University
27 Content: Orange carrots are packed with chemicals called carotenoids-specifically, beta-carotene. Your body
        turns beta-carotene into vitamin A, which helps you grow and protects you from getting sick.
        Beta-carotene isn't just nutritious. It's also loaded with orange pigment.
28 That's why vegetables with lots of beta-carotene-like sweet potatoes, squash, and pumpkins-share the same
        color. But what about that rainbow of other carrot colors? They have their own special qualities, too.
        Purple carrots get their color from
29 ]
30 Output: The query asks about chemical that makes carrot orange. Because there's one high confidence result,
        the most relevant knowledge entries about such chemical is "High Confidence Response: carotenoids."
31 From this result we know the chemical shall be carotene.
32 Therefore, the predicted answer is carotene.
33
34
35
36 Query: What is the name of the drainage basin of ounasjoki?
37 Knowledge: [
38 Title: Ounasjoki - Wikipedia
```

39 Content: It is also the largest river entirely within its borders. Ounasjoki is approximately 299.6
    kilometres (186.2 mi) in length, and the catchment area is 13,968 square kilometres (5,393 sq mi), 27%
    of the Kemijoki catchment area.
40 Tributaries
41
42 - Nakkalajoki.
43 - Kakkalojoki.
44 - Syva Tepastojoki.
45 - Loukinen.
46 - Meltausjoki.
47 Course. The Ounasjoki originates at Ounasjarvi lake in Enontekio. It flows first eastwards through
    Perilajarvi lake and turns south after some seven kilometres. The river then follows southern-sou
48
49 Title: DRAINAGE BASIN OF THE BALTIC SEA - UNECE
50 Content: Vistula. 194,424. Baltic Sea. BY, PL, SK, UA. - Bug. 39,400. Vistula. BY, PL, UA. - Dunajec. 4726.7.
    Vistula. PL, SK. -Poprad. 2,077. Dunajec. PL, SK. Oder. 118,861. Baltic Sea. CZ, DE, PL. - Neisse ...
    Oder. CZ, DE, PL. - Olse ... Oder. CZ, PL. 1 The assessment of water bodies in italics was not included
    in the present publication. 2 For the Venta River Basin District, which includes the basins of the
    Barta/Bartuva and Sventoji rivers. Oulu. Lulea. Rovaniemi. Lake. Oulujarvi. Lake. Tornetrask. Torne.
    Oulujoki.
51 ]
52 Output: The query asks about drainage basin of ounasjoki. The most relevant knowledge entry that contain
    basin is "Venta River Basin District, which includes the basins of the Barta/Bartuva and Sventoji
    rivers."
53 From this result we know the drainage basin shall be Venta River Basin.
54 Therefore, the predicted answer is Venta River Basin.
55
56
57 Query: What is the typical diameter (in centimetre) of tennis?
58 Knowledge: [
59 Title: What Size Is A Tennis Ball In Cm? - Metro League
60 Content: To Recap. A tennis ball is typically about 2 cm in diameter. Similar Posts: What Is A Junk Ball In
    Tennis?
61 How tall is a tennis ball? Tennis Balls come in different sizes, some as small as 2.575"-2.7" (6.54-6.86 cm)
    and others up to 8 inches (20 cm). The mass of a tennis ball must be between 1.975-2.095 oz (56-59 g).
62
63 Title: Tennis Ball Dimensions & Drawings | Dimensions.com
64 Content: Tennis Balls have a diameter of 2.575"-2.7" (6.54-6.86 cm) and circumference of 8.09"-8.48"
    (20.6-21.5 cm). The mass of a Tennis Ball must be between 1.975-2.095 oz (56-59.4 g).
65 Tennis Balls have a diameter of 2.575"-2.7" (6.54-6.86 cm) and circumference of 8.09"-8.48" (20.6-21.5 cm).
    The mass of a Tennis Ball must be between 1.975-2.095 oz (56-59.4 g). A Tennis Ball is a ball designed
    for the sport of tennis.
66
67 Title: Tennis ball - Wikipedia
68 Content: Modern tennis balls must conform to certain criteria for size, weight, deformation, and bounce
    criteria to be approved for regulation play. The International Tennis Federation (ITF) defines the
    official diameter as 6.54-6.86 cm (2.57-2.70 inches). Balls must have masses in the range 56.0-59.4 g
    (1.98-2.10 ounces).
69 ]
70 Output: The query asks about diameter of tennis (in centimetre). the most relevant knowledge entry about
    diameter of tennis is "tennis balls have a diameter of 2.575"-2.7" (6.54-6.86 cm) and circumference of
    8.09"-8.48" (20.6-21.5 cm)".
71 As the query ask about centimetre, cm. From this result we know the diameter shall be 6.54 - 6.86.
72 Therefore, the predicted answer is 6.54 - 6.86.
73
74
75
76 Query: Who is the inventor of women netball, sport?
77 Knowledge: [
78 Title:
79 Content: History of netball - Wikipedia
80
81 In 1893, Martina Bergman-osterberg informally introduced one version of basketball to her female physical
    training students at the Hampstead Physical Training College in London, after having seen the game
    being played in the United States.
82
83 Title: History of netball - Wikipedia
84 Content: In 1893, Martina Bergman-osterberg informally introduced one version of basketball to her female
    physical training students at the Hampstead Physical Training College in London, after having seen the
    game being played in the United States. Madame osterberg advocated physical fitness for women to better
    prepare them for motherhood and in the wider context of women's emancipation.
85
86 Title: Netball - Wikipedia
87 Content: A common misunderstanding of netball's origins has resulted in the mistaken belief that netball was
    created to prevent women from playing basketball. However, netball's development traces back to
    American sports teacher Clara Gregory Baer's misinterpretation of the basketball rule book in 1895.
88 History. Netball's early development emerged from Clara Baer's misinterpretation of the early rules of James
    Naismith's new sport of basketball (which he developed while studying in Massachusetts) and eventually
    evol
89
90 Title: History of Netball - World Netball
91 Content: Women's indoor basketball began exactly two days later when female teachers to the gym were
    captivated by the game but it wasn't until 1895 that the current game of netball was well and truly
    shaped. When Clara Baer, a sports teacher in New Orleans, wrote to Naismith asking for a copy of the
    rules, the subsequent rules package contained a drawing of
92 ]
93 Output: The query asks about inventor of women netball. The most relevant knowledge entry about women netball
    inventor is "In 1893, Martina Bergman-Osterberg informally introduced one version of basketball to her
    female physical training students".
94 From the result, we know the inventor shall be Martina Bergman-Osterberg.
95 Therefore, the predicted answer is Martina Bergman-Osterberg.
96

```
97
98 Query: How many elevators does torre picasso have?
99 Knowledge: [
100 Title:
101 Content: Torre Picasso | Turismo Madrid
102
103 The interior of the Picasso Tower houses offices designed as intelligent spaces equipped with the highest
      technology, comfort and use of space. It has 18 lifts, divided into three groups of six.
104
105 Title: Torre Picasso - Wikipedia
106 Content: 26 elevators; 18 serve office floors divided into three zones:
107
108 - 1st-18th floors at 2.5 m/s (8.20 ft/s)
109 - 18th-32nd floors at 4 m/s (13.12 ft/s)
110 - 32nd-43rd floors at 6 m/s (19.69 ft/s) (fastest in Spain)
111
112 Title: Torre Picasso - Field Trip
113 Content: 26 elevators, of which 18 to office floors in 3 groups of 6:
114
115 - 1st-18th floors at 2.5 m/s (8.20 ft/s)
116 - 18th-32nd floors at 4 m/s (13.12 ft/s)
117 - 32nd-43rd floors at 6 m/s (19.69 ft/s) (apparently fastest in Spain)
118
119 Title: Torre Picasso - Wikiwand
120 Content: The building as seen from the junction of the Paseo de la Castellana and the Plaza de Pablo Ruiz
      Picasso. 26 elevators; 18 serve office floors divided into three zones: 1st-18th floors at 2.5 m/s
      (8.20 ft/s) 18th-32nd floors at 4 m/s (13.12 ft/s)
121
122 ]
123 Output: The query asks about number of elevators in torre picasso. the most relevant knowledge entry about
      number of elevators in torre picasso is "26 elevators; 18 serve office floors divided into three
      zones:".
124 From the result, we know the number of elevators shall be 26.
125 therefore, the predicted answer is 26.
126 """
```

Listing 6: Reason Prompt (Knowledge Question)

```python
1 class MemoryState:
2   state: str = ''
3   traversed_actions: list = []
4   query: str = ''
5   context: str = ''
6
7   def __init__(self, state, query = '', context = ''):
8     self.state = state
9     self.query = query
10    self.context = context
11
12 def plan(transition_graph, cur_memory, lens_res, retr_res):
13   action_list = [a for a in transition_graph[cur_memory.state] if a not in cur_memory.traversed_actions]
14   action_prompt = ''
15   for a in action_list:
16     action_prompt += '  --' + a + ': ' + task_instructions[a] + '\n'
17   prompt_example = ""
18   for a in action_list:
19     prompt_example += action_prompt_dict[a] + "\n"
20   action_prompt = planner_prompt % (cur_memory.query, action_prompt, prompt_example, cur_memory.query,
      cur_memory.context)
21   action = api_utils.call_palm(action_prompt)[0]
22
23   instruction = []
24   if action in require_instruction:
25     exclude_ids = cur_memory.traversed_actions:
26     prompt = instruction_prompt(cur_memory.query, lens_res, exclude_ids)
27     res = api_utils.call_palm(prompt)[0]
28     reason = parse_reason('the query asks about ' + reason)
29     instruction = [reason, res]
30   return action, instruction
31
32 def avis_execution(d):
33   state = 'START'
34   answer = None
35
36   prompt = question_decomposition_prompt + 'Question: ' + q + '\n'
37   res = api_utils.call_palm(prompt)[0]
38
39   vqi = res.find('Visual: ')
40   kqi = res.find('Knowledge: ')
41   vq = res[vqi + 8: kqi-1]
42   kq = res[kqi+11:]
43
44   working_memory = [MemoryState(state = 'START', query = vq, context = lens_res[0])]
45   while not answer:
46     cur_memory = working_memory[-1]
47     action, instruction = plan(transition_graph, cur_memory, lens_res, retr_res)
48     exec_res = execute(action, instruction, lens_res, retr_res)
49     res = reason(exec_res)
50     if 'not informative' in res:
51       cur_memory.traversed_actions += [action]
52     elif 'answer is' in res:
```

```
53        answer = res[10:]
54     else:
55        working_memory += [MemoryState(state = action, query = kq, context = res)]
56  return answer
```

Listing 7: Workflow of AVIS (code snippets)

# F   Error Analysis

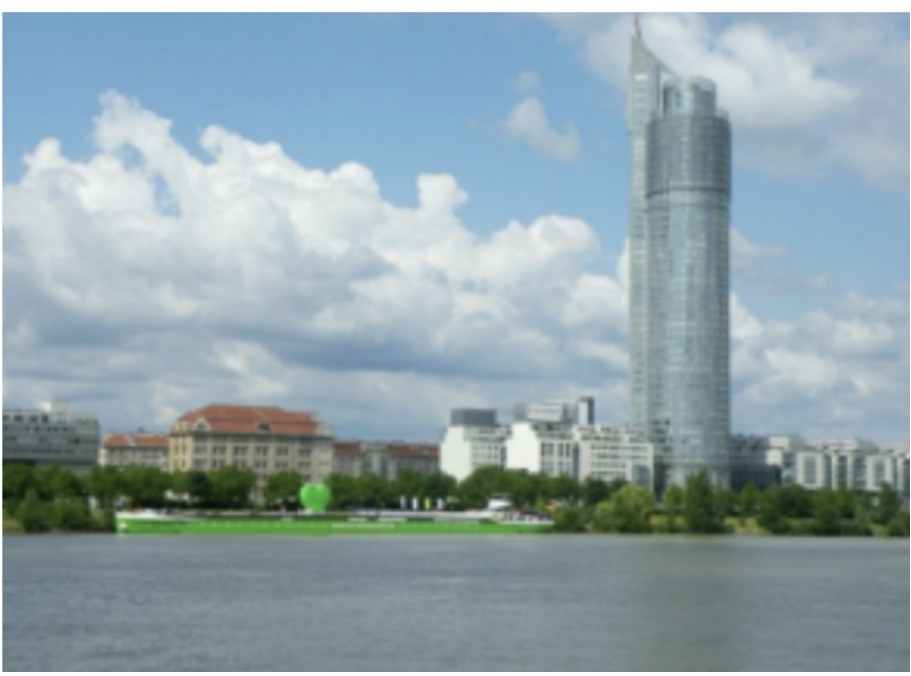

Figure 14: **Error Type**: LLM planning module miss important details. **Qustion**: What country does this building belong to? **Correct answer**: Austria; **Predicted answer**: United States

- Step 1: object_select, query = What is this building?
- Step 2: image_search, query = what is this building?

  Retrievals: [ Millennium tower wien hi-res stock photography and images - Alamy (score=48.4), Millennium Tower Office Building in Vienna, Austria......

  Reason: The query asks about the building. From Object, there is a mention about "Millennium Tower" with high score. From Retrievals, there are many mentions about "Millennium Tower" with high score. Therefore, the predicted answer is Millennium Tower, building.

- Step3: Google Search, query = country of Millennium Tower, building?

  Title: Number 8: Millennium Tower in SoMa, San Francisco - San Francisco YIMBY Content: The eighth tallest skyscraper in the Bay Area planned or built is the Millennium Tower...

  From the search result, "The Millennium Tower is a skyscraper in San Francisco, California", the predicted answer is United States.

  yes, answerable, Output = United States

- image_search query=what is this bridge?

  Retrievals: high confidence: Na Drini ćuprija - Zadužbina Mehmed-paše Sokolovića - Furaj.ba | S nama u avanturu (score=87.2), high confidence: File:NKD279 Most Mehmed Paše Sokolovića Višegrad 5.jpg - Wikimedia Commons (score=91.3), ......]

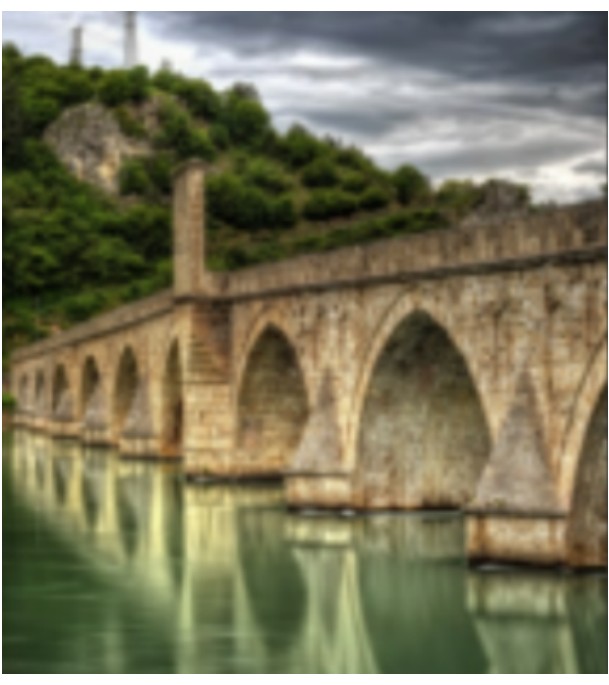

Figure 15: **Error Type**: LLM QA module extracts wrong evidence. **Qustion**: What is the architectural style of this bridge? **Correct answer**: Ottoman architecture; **Predicted answer**: turkish monumental architecture

yes, answerable. The query asks about the bridge. From Retrievals, there are many mentionings about "Mehmed Paša Sokolović Bridge". Therefore, the predicted answer is Mehmed Paša Sokolović Bridge, bridge.

- Google Search, query=architectural style of Mehmed Paša Sokolović Bridge, bridge?

  Title: Mehmed Pasha Sokolovic Bridge – Višegrad, Bosnia and Herzegovina - Atlas Obscura Content: Celebrated example of Ottoman architecture that also served as a site for brutal massacres. Been Here? ......

  yes, answerable. Question asks about architectural style of Mehmed Paša Sokolović Bridge, bridge. From the search resutls, the most relevant knowledge entry about architectural style is "It is characteristic of the apogee of Turkish monumental architecture and civil engineering". Therefore, the predicted answer is Turkish monumental architecture.

- Step 1: identical_image_search, query = What is this place? Retrievals: [ high confidence: Reportage entre les murs de Mariama Bâ, internat d'excellence pour jeunes filles sénégalaises - Debout Congolaises (score=95.3), ] not informative, **backtrack**

- Step 2: object_select, query = What is this place? the query asks about the place. the image contains boats, water, trees and buildings. there is no explicit information about the place. therefore, the predicted object id is 0, which is whole image.

  Step 3: image_search, query = What is this place? Object: [ a group of boats floating on top of a body of water . There are two white boats on the water. Behind that there are trees and buildings. (Caption, whole image) ] Retrievals: [ high confidence: Reportage entre les murs de Mariama Bâ, internat d'excellence pour jeunes filles sénégalaises - Debout Congolaises (score=95.3), ] not informative, **backtrack**

- Step 4: vqa, query = What is this place? output=a lake yes, answerable

- Step 5: llm_qa, query = body of water is a lake located in or next to? output=land

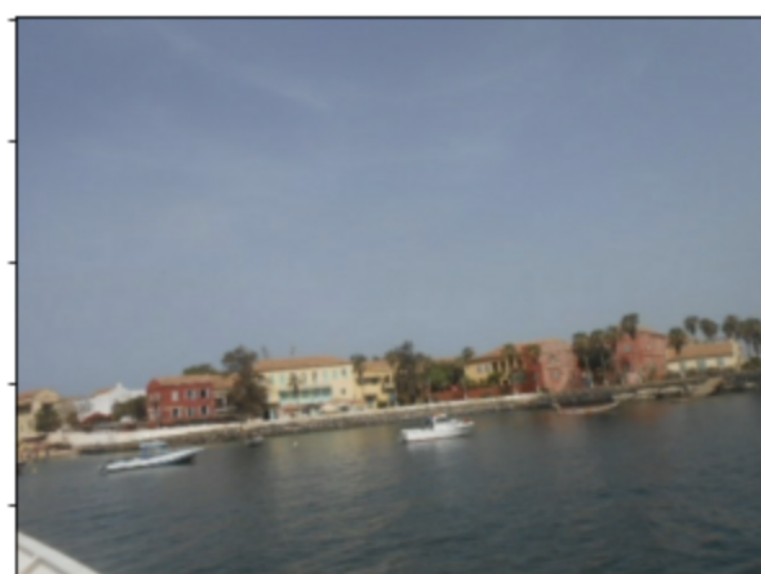

Figure 16: **Error Type**: Tool provides incorrect information. **Qustion**: Which body of water is this place located in or next to? **Correct answer**: Atlantic; **Predicted answer**: land