# OpenReview forum: "AVIS: Autonomous Visual Information Seeking with Large Language Model Agent"
_NeurIPS.cc/2023/Conference — NeurIPS 2023 poster_

### Official Review · Reviewer_M3n5 · 2023-06-30

**Soundness:** 3 good
**Presentation:** 3 good
**Contribution:** 3 good
**Rating:** 6
**Confidence:** 4

**Summary:**

1. This paper proposes a novel VQA framework that uses LLMs to select external tools in multiple stages to extract the necessary information to answer the question.
2. The authors gather human decision data to develop a decision graph and construct in-context examples to guide the LLM to perform API selection and query construction.
3. The experiments validate the effectiveness of the proposed method on two knowledge-based VQA benchmarks.

**Strengths:**

1. The experiment result is solid and shows the advantage of the framework and the improvement of each sub-module.
2. The guidance for tool selection from human demonstration is novel, and the huge improvement also gives us some insight into the weakness of LLMs.

**Weaknesses:**

1. The LLM used is PALM, while the compared methods used GPT series models. It's better to have some results based on GPT models to demonstrate the effectiveness of the proposed framework.
2. The proposed human guidance requires a certain amount of human labor and may be hard and costly to generalize to other tasks using LLMs (manipulation, navigation, etc.).


**Questions:**

1. It will be interesting to see how the framework performs on zero-shot setting and related discussions, especially the related work ViperGPT is zero-shot.

[1] ViperGPT: Visual Inference via Python Execution for Reasoning

**Limitations:**

The limitation and potential solutions are discussed.

---

> ### Author Rebuttal · Authors · 2023-08-10
>
> **Q. Use GPT**
>
> We change the backbone LLM as GPT4
>
> | Model Configuration                   | Result (%) |
> |--------------------------------------|------------|
> | AVIS w/ GPT-4                        | 61.9       |
> | GPT-4 w/ PALI*                       | 13.1       |
> | GPT-4 w/ PALI* + Object              | 36.4       |
> | GPT-4 w/ PALI* + Object + Search     | 43.8       |
>
> As it shows, AVIS’s decision framework can still benefit a powerful LLM as GPT-4, showing its generalizability to different LLM.
>
>
> **Q: The proposed human guidance requires a certain amount of human labor and may be hard and costly to generalize to other tasks using LLMs**
>
> A: For AVIS, we carried out a user study that engaged 10 participants, who collaboratively responded to a cumulative count of 644 visual inquiries. On average, an individual took approximately 1 minute to address a visual question utilizing our user interface. As a result, we expended roughly 10 hours of human effort on AVIS. The planner and reasoner, grounded in LLM technology, demonstrated adeptness in generalization with minimal exposure to contextual instances. This characteristic substantially diminishes the demand for human labor.
>
> A future work is to combine human supervision with reinforcement learning to further reduce the need for human annotations.
>
> **Q: Could AVIS with same prompt generalize to other tasks?**
>
> We appreciate the emphasis on the generalizability and practicality of our model. To address this concern, we've conducted experiments with the A-OKVQA dataset, which, share similar question types as OK-VQA and Infoseek but have unique properties (as discussed in their paper). Importantly, we adopted the same prompts used in Infoseek and OKVQA for our AVIS evaluation on A-OKVQA, without the need for additional human annotations.
>
> | Model Configuration                      | Result (%) |
> |------------------------------------------|------------|
> | AVIS                                     | 56.7       |
> | PALM + PALI*                             | 41.6       |
> | PALM + PALI* + Object                    | 47.6       |
> | PALM + PALI* + Object + Search           | 50.8       |
> | GPV-2                                    | 48.6       |
> | KRISP                                    | 33.7       |
>
> This indicates the generalizability of our framework with the great reasoning capability of existing LLM, which shall be able to transfer the decision capability across datasets, and reduce the requirement to annotate additional prompts for some new VQA datasets.
>
> For other types of tasks reviewer mention (e.g, manipulation, navigation), we definitely need another set of prompts as their underlying logic is very different from VQA. But similar as what we observe in VQA, we believe that AVIS with a designed transition graph of a certain task shall have some sorts of transferability and generalization across different environments and distributions.
>
>
> **Q: Zero(one)-shot setup**
>
> A: We'd like to emphasize that ViperGPT is not zero-shot, it provides one example in its documentation per API. To make a fair comparison, we also keep one prompt for each decision action, and don't use any in-context example (zero-shot) for reasoning, and the result on OK-VQA is 53.2, slightly higher than ViperGPT which is 51.9. Note that the two framework doesn't use the same set of API, so it's not directly comparable. But it shows that our framework AVIS could also work for cases without many prompts.

---

> > ### Comment · Reviewer_M3n5 · 2023-08-17
> >
> > Thanks to the authors for the response!
> > For GPT models, I intended to see how the framework compares with ViperGPT (using GPT-3). Therefore, an experiment with GPT-3 (or 3.5) will reflect how the proposed framework generalized to weaker LLMs (as GPT4 and Palm 540B could be seen as the most powerful LLMs by far) and compare with ViperGPT. From my understanding, the prompt and dialog history in the proposed framework is long. Therefore, I'm not sure if weaker LLMs can follow it.

---

> > > ### Author Response · Authors · 2023-08-17
> > > **Respond**
> > >
> > > Thanks so much for the clarification. We don't have ViperGPT's results on Infoseek dataset, so we run our framework with ChatGPT (GPT3.5) on OK-VQA dataset
> > >
> > > Model Configuration  |       Accuracy on OKVQA (%)
> > > --- | -----------
> > > AVIS w/ GPT-3.5         |      61.1
> > > AVIS w/ GPT-3.5    (one-shot)     |      53.9
> > > GPT3.5 w/ PALI*          |     47.2
> > > GPT3.5 w/ PALI* + Object.  | 42.6
> > > GPT3.5 w/ PALI* + Object + Search   |50.3
> > >
> > > The results show that AVIS could also benefit GPT3.5, and the final result (both one-shot & full) is higher than what is reported in ViperGPT (51.9)
> > > Note that we doesn't aim to do a rigorous comparison to ViperGPT, as the tools used are different. As LLM Agent is still a vibrant field, how to unifying different existing methods into a same setup is a ongoing challenge. We're also reproducing ViperGPT in our setting, and we plan to add the results of it with our tool in later version.
> > >
> > > And We're also interested to see whether our dynamic decision making w/ human defined transition graph (or we could call it domain-specific language, DSL) could also benefit program synthesis like ViperGPT, which we leave for future work.
> > >
> > >
> > >
> > > Another thing I'd like to clarify. Although we store all the past dialog history into the working memory, not all of them are used to serves as prompt to the LLM. Instead, only the results related to the current state is used to fill in the prompt defined by human examples. The purpose of it is to reduce the inference cost when making the decision (as we need to run LLM multiple times to solve a query, if the last few decision accumulates history then the computational cost is very large, and the context length could even be larger than the largest window size current LLM could take). On the contrary, currently each decision making at arbitrary step cost similar.
> > >
> > > You could also look at our discussion with Reviewer MSpf about a similar question on "The inclusion of uninformative outputs in working memory may potentially impact the obtained results". Hope this could clarify our framework better.

---

### Official Review · Reviewer_hDYr · 2023-07-01

**Soundness:** 4 excellent
**Presentation:** 3 good
**Contribution:** 3 good
**Rating:** 7
**Confidence:** 4

**Summary:**

This paper proposes a VQA system that mimics the human decision-making process and leverages LLM and web searches to perform multimodal reasoning. The system consists of three main components: a transition graph, an LLM planner, and an LLM Reasoner. The transition graph is manually designed based on the human decision-making process for knowledge-intensive VQA. It guides the system to infer the answer from different states of information. The transition graph enhances the interpretability of the inference process and allows the system to focus on the current state only. The transition graph also introduces uncertainty to the decision procedure, making it more dynamic than previous methods that use static plans for tool usage. The LLM planner decides the next action (API calls and query content) based on the current state. The LLM Reasoner produces the answers from the collected information. Experiments demonstrate the effectiveness of this method on Visual-Question answering tasks OK-VQA and Infoseek.

**Strengths:**

Leveraging the user study to build a state transition graph is a reasonable way to construct LLM applications, especially for well-understand applications such as VQA or specific expert domains.

The motivation is clear and the writing is easy to follow.

Experiments are well-designed to show the effectiveness of the method.

**Weaknesses:**

Lack of error analysis.

**Questions:**

How does the system handle the situation when the output is not informative and the pipeline gets stuck in an infinite loop?

Regarding the weakness: The model improved the performance on the dataset, but it is still far from perfect. It would be helpful to analyze the sources of the errors (Prompts? LLM? transition graph?) which will benefit the community.

**Limitations:**

This paper acknowledges the limitations of the model: 1. it is computationally intensive, and 2. it is currently only suitable for the VQA task.

negative societal impact: not found

---

> ### Author Rebuttal · Authors · 2023-08-10
>
> **Q: Error Analysis**
>
> A: We look through the error samples by AVIS and categorize them  with three major types of error:
>
> - LLM Planning Module Errors: Cases where the LLM planning component failed to discern crucial information, leading to inaccurate decision-making.
>
> - LLM Reasoning (QA) Module Errors: Instances where the LLM's reasoning and question-answering capabilities identified incorrect evidence or derived flawed inferences.
>
> - Tool-Induced Errors: Situations where the external tools, invoked by the AVIS framework, furnished misleading or incorrect information.
>
> To provide a more tangible understanding of these errors, we've detailed specific instances of each type in an attached one-page PDF file, offering a granular view of the issues encountered.
>
> To further contextualize, we carried out an analysis on 30 randomly chosen misclassified samples from the Infoseek dataset. Our findings are summarized as follows:
>
> - Reasoning & QA Errors: 11 instances (36.7%)
> - Tool-Induced Errors: 7 instances (23.3%)
> - LLM Planning Module Errors: 6 instances (20%)
> - Ambiguous or Imperfect Matches: For the remaining 6 instances (20%), our assessment suggests that AVIS's prediction was accurate. The classification as erroneous likely arises due to slight mismatches or semantic nuances.
>
>
> By far we haven't found examples that are caused by incompleteness of transition graph, as we think it already cover most of the possible and promising actions, and the mistakes are mainly caused by LLM didn't make correct decision. Probably when the number of tools increase, the error caused by transition graph will be more prominent, which we leave such analysis for future work.
>
>
> **Q.  Will AVIS be stuck into an infinite-loop as AutoGPT?**
>
> A: Unlike models such as AutoGPT, AVIS has been designed with specific safeguards against infinite-loop scenarios:
>
> Transition graph: At the heart of our model lies a transition graph defining all possible actions for executing tools. This ensures that our search space remains finite (every action our planner of AVIS generates is a <tool_id, input_arg> pair). This is very different from some other autonomous agents such as AutoGPT that use LLM’s generated language as action, which have infinite search space.
> Traversal history & repetition removal: As AVIS navigates over a given transition graph with finite paths, we could easily record all previous traversed states (similar to standard DFS algorithm). In this way, everytime we make the planning, we will remove those traversed states and only ask the model to predict the actions that have not traversed before, which avoids redundancy & potential infinite-loop.
> Terminating Decision: Based on the design, if AVIS tries to traverse all possible paths and still cannot find the answer, AVIS will terminate and output “we cannot find the answer”, instead of falling into an infinite-loop.
>
> Through these design strategies, we've ensured that AVIS remains efficient in its operations without getting ensnared in unproductive loops.

---

> > ### Comment · Reviewer_hDYr · 2023-08-15
> >
> > Thank you for the response. I have raised the "soundness" score to 4, and I would like to maintain my overall rating i.e., I still recommend acceptance for this work.

---

> > > ### Author Response · Authors · 2023-08-16
> > > **Thanks for your reviews and responses**
> > >
> > > Thanks so much for your recognition of our work. We will definitely include the error analysis and the discussion about how we handle infinite-loop into the paper.

---

### Official Review · Reviewer_MSpf · 2023-07-03

**Soundness:** 3 good
**Presentation:** 3 good
**Contribution:** 3 good
**Rating:** 6
**Confidence:** 3

**Summary:**

This paper introduces an autonomous framework for visual question answering framework named AVIS. AVIS utilizes a Large Language Model (LLM) as its core component to dynamically strategize the utilization of external tools. The framework comprises three key components: the planner, reasoner, and working memory. The planner determines the appropriate actions, such as selecting relevant APIs, in each step. The working memory stores the results obtained from executing the APIs, while the reasoner processes the outputs derived from the API calls. Experimental evaluations conducted on the OK-VQA and Infoseek datasets validate the effectiveness of the AVIS framework.

**Strengths:**

1. The authors present a novel framework that leverages external tools to overcome the challenge of external knowledge dependency in Visual Question Answering (VQA) tasks, thereby enhancing the practicality of VQA in real-world scenarios.

2. The authors introduce a dynamic decision process for selecting the most suitable external APIs to address specific sub-problems at each step.

3. The authors utilize human decision-making data to construct a transition graph, which is helpful for narrowing down the action spaces in the decision-making process.

4. Extensive experiments on the OK-VQA and Infoseek datasets demonstrate the effectiveness of AVIS.


**Weaknesses:**

1. AVIS necessitates the utilization of costly human-annotated data to guide the (LLM). To further substantiate the generalizability and practicality of AVIS, it would be advantageous to conduct experiments on the A-OKVQA[1] dataset, which incorporates the OK-VQA and Infoseek human-annotated data as in-context samples.

2. As indicated in Line 5 of Algorithm 2, the authors directly incorporate the outputs into the working memory. However, the inclusion of uninformative outputs may potentially impact the obtained results.

3. For the three baseline methods, it would be better to demonstrate the few-shot COT prompts.

4. In Line 250, some human-annotated samples may be “Could not find the answer”. How do the authors leverage this type of sample? When using this type of sample as an in-context sample, does it lead to the models rejecting the question when generating a response?

5. It would be advantageous to provide an average step count that AVIS typically requires to address a single VQA sample.

6. In Line 300, “Table 4” should be “Table 1”.


**Questions:**

Refer to Weakness

---

> ### Author Rebuttal · Authors · 2023-08-10
>
>
> **Q: it would be advantageous to conduct experiments on the A-OKVQA[1] dataset, which incorporates the OK-VQA and Infoseek human-annotated data as in-context samples.**
>
> A: We appreciate the emphasis on the generalizability and practicality of our model. To address this concern, we've conducted experiments with the A-OKVQA dataset, which, share similar question types as OK-VQA and Infoseek but have unique properties (as discussed in their paper). Importantly, we adopted the same prompts used in Infoseek and OKVQA for our AVIS evaluation on A-OKVQA, without the need for additional human annotations.
>
> | Model Configuration                      | Result (%) |
> |------------------------------------------|------------|
> | AVIS                                     | 56.7       |
> | PALM + PALI*                             | 41.6       |
> | PALM + PALI* + Object                    | 47.6       |
> | PALM + PALI* + Object + Search           | 50.8       |
> | GPV-2                                    | 48.6       |
> | KRISP                                    | 33.7       |
>
> This indicates the generalizability of our framework with the great reasoning capability of existing LLM, which shall be able to transfer the decision capability across datasets, and reduce the requirement to annotate additional prompts for some new VQA datasets.
>
>
> **Q. The inclusion of uninformative outputs in working memory may potentially impact the obtained results**
>
> A. Our approach to handling failure cases within the working memory is two-pronged:
>
> Differentiation between working memory and LLM prompt:
> Firstly, it's essential to clarify that the working memory of AVIS is distinct from the prompt provided to the LLM. Within the working memory, we do maintain a record of both successful and unsuccessful queries. However, when interfacing with the LLM, only the most pertinent or top-ranked information is presented, ensuring that the LLM isn't unduly influenced by extraneous or unproductive details.
>
> Purpose of retaining uninformative records:
> Our goal of recording the uninformative path (failure) is to avoid repetition of previously traversed paths. When we make the planning, we only provide the model the actions that it has not traversed before, and remove all uninformative actions stored in working memory. This helps AVIS avoid going to the same uninformative path again and solves the repetitive and even infinite loops.
>
>
> **Q: How are the examples where human indicates “cannot find answer” are used to guide AVIS?**
>
> A: The inclusion of "cannot find answer" human annotations serves a strategic purpose in AVIS's decision-making mechanism:
>
> Incorporation into the prompt: We integrate these specific annotations into the model's prompt, ensuring that the model is aware of scenarios where human annotators couldn't discern a clear answer. This awareness is pivotal in providing a holistic context to the LLM during its reasoning process.
>
> Avoid hallucination: Should AVIS exhaust all action spaces and still be unable to generate a result with high confidence, these "cannot find answer" annotations guide it to echo a similar response. In practice, the frequency of such outcomes is within an acceptable range. By mirroring human annotator behavior in these challenging scenarios, we aim to provide a realistic and candid response to users, rather than forcing a potentially inaccurate answer (i.e., hallucination).
>
>
> **Q: Average step count**
>
> A: The average step of AVIS is 5.2, and the full distribution is shown in Fig 8 of Appendix.
>
> **Q: In Line 300, “Table 4” should be “Table 1”.**
>
> A: Thank you for pointing out this typo. We will fix it.
>
>
> **Q: Show COT prompts for other baselines**
>
> A: Sure, we will definitely include all the details in appendix as well as open-sourced codes. Below we show a few examples:
>
>
>
> pali_prompt = """
>
> Please based on the "Caption" to answer the question:
>
>
> Question: What type of fruit are they holding?
> Caption: a couple of men standing next to each other holding oranges . There are two persons standing and holding oranges in their hands and there are few people beside them and there is a building in the background.
> reason: from caption, it says the men are holding oranges.
> Answer: orange
>
>
> ......
>
> """
>
>
> pali_object_prompt = """
>
> Please based on the "Caption" and "Entity" to answer the question:
>
>
> Question: What does the train carry?
> Caption: a train traveling down train tracks next to a forest . There are four trains on the railway track. In the background there are trees,poles and sky.
> Entity: [
> BNSF Railway: BNSF Railway is one of the largest freight railroads in North America (score=89.3)
> Extracted Text: BNSF (score=100.0)
> ]
> Reason: from caption, there are no enough information about what the train is carrying. From entity, it says the train is BNSF railway that is freight ralroads. So the train carry freight, which is good.
> Answer: good
>
> ......
>
>
> """
>
>
> For the PALI+Object+Search baseline, we adopt two-level procedure: first use pali_object_prompt to get visual answer, and then feed it into search API to get documents, with the same prompt shown in appendix to get final answer.

---

> > ### Comment · Reviewer_MSpf · 2023-08-16
> > **Response to Rebuttal**
> >
> > Thank you for the response. I have raised the "rating" score to 6.

---

> > > ### Author Response · Authors · 2023-08-16
> > > **Thanks for the response**
> > >
> > > Thanks so much for the review and response. The generalization study is definitely very important for LLM Agent, and we will include all the updated experiments in the paper.

---

### Official Review · Reviewer_mb9w · 2023-07-06

**Soundness:** 3 good
**Presentation:** 3 good
**Contribution:** 3 good
**Rating:** 4
**Confidence:** 4

**Summary:**

This work aims to more general VQA task (often needs external knowledge) via LLM-based information seeking. First, the info-seeking system is built with three components: planner, reasoner, and memory, seeking useful information with external tools/APIs. Second, they build a dataset with human decision-action samples via user study. Then, they build the transition graph from the dataset, for depicting the user state and action, which can be used as contextual instances serving the info-seeking system.

**Strengths:**

The info-seeking system is designed in the reasonable and mainstream VQA/QA fashion.

**Weaknesses:**

How to compare this work with the end-to-end frameworks? As LLMs growing gradually, along with the visual knowledges being grounded better and better with language knowledges, the external knowledges will become thinner and an inclusion of internal ones.
In general, this is a good paper with much groundwork. I am looking forward author feedback and better version to persuade me.


**Questions:**

How to avoid potential infinite loop such like those emerge in AutoGPT? Considering, rather than powerful LLMs, the capacity bottleneck is often caused by external tools.

**Limitations:**

None.

---

> ### Author Rebuttal · Authors · 2023-08-10
>
> Thank you for your valuable comments and remarks.
>
> **Q: Compare AVIS with end-to-end models? Will external knowledge still be useful?**
>
> A: The key difference between AVIS and a single end-to-end model is that AVIS separates knowledge memorization from reasoning. Within this architecture:
>
> - The external tools take the responsibility of memorizing diverse and flexible knowledge.
>
> - The neural network, freed from the necessity to store vast amounts of information, focuses its full capacity on planning and reasoning.
>
> Based on the modular nature of AVIS, adding or modifying knowledge doesn't necessitate finetuning any model parameters (which is costly especially for LLMs). Instead, one only needs to update the knowledge base in different tools (e.g., Google Search index). This provides distinct advantages in situations where end-to-end models might fail:
>
> - Handling addition of new knowledge (e.g., news & updates): While it's true that LLMs are able to memorize more and more facts within the model weights, it's essential to highlight that their knowledge is static and limited to the last training set. AVIS, through its dynamic utilization of APIs, can access real-time information and news updates that an LLM could not inherently possess. This ensures our model stays up-to-date with the latest information, which is crucial for many visual questions rooted in real-time contexts.
>
> - Domain-specific Knowledge: Despite the broad spectrum of information LLMs might encompass, specialized and long-tail domains often call for specific tools and databases. AVIS is primed for such integrations, making it a good candidate for domain-centric visual question-answering tasks.
>
> - Personal & private knowledge: Given privacy constraints, in many cases private data (such as photos & message history in personal phone) cannot be included into the training corpus of LLM, making it hard to become a personalized model. AVIS, on the other hand, can be customized to incorporate personal and private knowledge sources, should users consent, allowing for a more tailored and precise response mechanism.
>
> In summary, while the capabilities of LLMs are indeed advancing, there remains a distinct advantage in combining their reasoning prowess with the dynamic, real-time, and specialized knowledge retrieval capabilities of external tools, such as Google Search Engine and LENS.
>
>
> **Q: Will AVIS be stuck into an infinite-loop as AutoGPT?**
>
> A: Unlike models such as AutoGPT, AVIS has been designed with specific safeguards against infinite-loop scenarios:
>
> Transition graph for defined action spaces: At the heart of our model lies a transition graph defining all possible actions for executing tools. This ensures that our search space remains finite (every action our planner of AVIS generates is a <tool_id, input_arg> pair). This is very different from some other autonomous agents such as AutoGPT that use LLM’s generated language as action, which have infinite search space.
> Traversal history & repetition removal: As AVIS navigates over a given transition graph with finite paths, we could easily record all previous traversed states (similar to standard DFS algorithm). In this way, everytime we call the planner to perform an action, we remove the traversed state and only ask the model to predict the actions that are not traversed before, which avoids redundancy & potential infinite-loop.
> Decision to stop: Based on the design, if AVIS tries to traverse all possible paths and still cannot find the answer, AVIS will terminate and output “we cannot find the answer”, instead of falling into an infinite-loop.
>
> Through these design strategies, we've ensured that AVIS remains efficient in its operations without getting ensnared in unproductive loops.
>
>
> **Q: Is Tool the bottleneck or LLM?**
>
> A: In our general response, we show a detailed error analysis statistics (examples of each type shown in one-page pdf file). Among the three major error types, tool error consists of 23.3%, while the other two types of error caused by LLM planning and reasoning add up to 56.7%. We admit that these existing tools are definitely not perfect (otherwise we could directly use them for solving the task), but a good LLM as an agent shall be learnt to use only the important information from imperfect tools to make correct decision, which shares similar insights like the classical boosting idea, while we use LLM as controller to make the ensemble.
>
> Based on this and what we describe above, we believe that the key bottleneck for building up better AI agent is 1) the basic capability of LLM itself; 2) a better algorithm and framework that enable LLM to utilize external tools to provide required knowledge, and it could focus on reasoning. We believe the planning & reasoning framework of AVIS over a human-defined transition graph is one of a good starting point of such framework.

---

> > ### Author Response · Authors · 2023-08-19
> > **Any further question for discussion?**
> >
> > Dear Reviewer:
> >
> > As the author-reviewer discussion period is coming to an end, we wonder whether our response (especially the usefulness of external knowledge) has addressed your concerns? Looking forward to the further discussion regarding any more questions.

---

### Author Rebuttal · Authors · 2023-08-10

# Response to Reviewers

We thank the reviewers for their valuable comments and remarks.

In the rebuttal, we mainly add:

**1. Experimental results with GPT4 on Infoseek dataset**

| Model Configuration                   | Result (%) |
|--------------------------------------|------------|
| AVIS w/ GPT-4                        | 61.9       |
| GPT-4 w/ PALI*                       | 13.1       |
| GPT-4 w/ PALI* + Object              | 36.4       |
| GPT-4 w/ PALI* + Object + Search     | 43.8       |

**2. Generalization Analysis**
   - Showing that our model with the same prompts written on infoseek and okvqa generalize also to a-okvqa without re-human annotation.

| Model Configuration                      | Result (%) |
|------------------------------------------|------------|
| AVIS                                     | 56.7       |
| PALM + PALI*                             | 41.6       |
| PALM + PALI* + Object                    | 47.6       |
| PALM + PALI* + Object + Search           | 50.8       |
| GPV-2                                    | 48.6       |
| KRISP                                    | 33.7       |

**3. Compare with ViperGPT over one-shot setup**
   - We'd like to emphasize that ViperGPT is not zero-shot, it provides one example in its documentation per API. To make a fair comparison, we also keep one prompt for each decision action, and don't use any in-context example (zero-shot) for reasoning, and the result on OK-VQA is 53.2, slightly higher than ViperGPT which is 51.9. Note that the two framework doesn't use the same set of API, so it's not directly comparable. But it shows that our framework AVIS could also work for cases without many prompts.

**4. Error Analysis**
   - Conduct thorough error analysis with three major types of error: 1) LLM planning module miss important information to make mistake decision; 2) LLM reasoning (QA) module extracts wrong evidence; 3) Tool provide incorrect information
   - **Note:** We put the detailed examples of error in the one-page PDF file, please check it.
   - We randomly select 30 wrong classified samples of infoseek, and find that reasoning & QA part consists of 11 instances (36.7%); tool error consists of 7 instances (23.3%); and LLM planning module takes 6 instances (20%). There are other 6 instances we think the model make correct prediction, and it's just because of not perfectly matched


In the following, we will respond each authors' concerns.

---

> ### Author Response · Authors · 2023-08-14
> **Any further questions about rebuttal?**
>
> Dear reviewers:
>
> Thanks again for proposing all the constructive comments and questions in your reviews. Please let us know whether you have any further questions or concerns regarding to our responses.

---

### Decision · Program_Chairs · 2023-09-21

**Decision:**

Accept (poster)

**Comment:**

Three of the four reviewers voted for acceptance of this submission and confirmed that the rebuttal has addressed their major concerns. Only one reviewer still remains negative after the rebuttal. The reviewer mb9w's major concern is how to compare this work with the end-to-end frameworks. The AC has checked the authors' response and believes the rebuttal is strong and sufficient. However, the reviewer mb9w has no responses in the discussion period so not sure whether he/she checked the authors' responses. The AC made the final acceptance recommendation and encouraged the authors to include all the discussion and responses in the final version.